# The RNA-binding protein SFPQ preserves long-intron splicing and regulates circRNA biogenesis in mammals

**Lotte Victoria Winther Stagsted, Eoghan Thomas O'Leary, Karoline Kragh Ebbesen, Thomas Birkballe Hansen***

Department of Molecular Biology and Genetics, Aarhus University, Aarhus, Denmark

**Abstract** Circular RNAs (circRNAs) represent an abundant and conserved entity of non-coding RNAs; however, the principles of biogenesis are currently not fully understood. Here, we identify two factors, splicing factor proline/glutamine rich (SFPQ) and non-POU domain-containing octamer-binding protein (NONO), to be enriched around circRNA loci. We observe a subclass of circRNAs, coined DALI circRNAs, with distal inverted *Alu* elements and long flanking introns to be highly deregulated upon SFPQ knockdown. Moreover, SFPQ depletion leads to increased intron retention with concomitant induction of cryptic splicing, premature transcription termination, and polyadenylation, particularly prevalent for long introns. Aberrant splicing in the upstream and downstream regions of circRNA producing exons are critical for shaping the circRNAome, and specifically, we identify missplicing in the immediate upstream region to be a conserved driver of circRNA biogenesis. Collectively, our data show that SFPQ plays an important role in maintaining intron integrity by ensuring accurate splicing of long introns, and disclose novel features governing *Alu*-independent circRNA production.

*For correspondence:
tbh@mbg.au.dk

**Competing interests:** The authors declare that no competing interests exist.

## Introduction

Gene expression is the output of multiple tightly coupled and controlled steps within the cell, which are highly regulated by a variety of factors and processes. Among these are the physical and functional interactions between the transcriptional and splicing machineries that are of great importance for the generation of both canonical and alternative isoforms of RNA transcripts. This includes a novel class of unique, closed circular RNA (circRNA) molecules.

CircRNAs are evolutionary conserved and display differential expression across cell types, tissues, and developmental stages. The highly stable circular conformation is obtained by covalently joining a downstream splice donor to an upstream splice acceptor, a backsplicing process catalyzed by the spliceosome (*Memczak et al., 2013*; *Jeck et al., 2013*; *Salzman et al., 2013*; *Ashwal-Fluss et al., 2014*; *Hansen et al., 2013*). The vast majority of circRNAs derive from coding sequences, making their biogenesis compete with the production of linear isoforms (*Ashwal-Fluss et al., 2014*; *Salzman et al., 2013*, *Salzman et al., 2012*). Complementary sequences in the flanking introns can facilitate the production of circRNAs (*Dubin et al., 1995*; *Ivanov et al., 2015*; *Jeck et al., 2013*; *Westholm et al., 2014*; *Zhang et al., 2014*), where the primate-specific *Alu* repeats are found to be significantly enriched in the flanking introns of circRNAs (*Jeck et al., 2013*; *Ivanov et al., 2015*; *Venø et al., 2015*). In some cases, exon skipping has been shown to stimulate circularization of the skipped exon (*Barrett et al., 2015*). However, in both human and *Drosophila*, biogenesis of the most abundant and conserved pool of circRNAs tend to be driven by long flanking introns rather than the presence of proximal inverted repeats in the flanking sequences (*Westholm et al., 2014*; *Stagsted et al., 2019*). The biogenesis of circRNAs without inverted repeats is currently not

understood in detail, although RNA-binding proteins (RBPs) associating with the flanking introns of circRNAs have been shown to be important (*Ashwal-Fluss et al., 2014*; *Conn et al., 2015*; *Errichelli et al., 2017*).

Here, we aim to identify additional protein factors involved in circRNA biogenesis. To this end, we exploited the enormous eCLIP and RNA sequencing resource available from the ENCODE consortium (*ENCODE Project Consortium, 2012*). Stratifying eCLIP hits across the genome with circRNA loci coordinates revealed the splicing factor proline/glutamine rich (SFPQ) and non-POU domain-containing octamer-binding protein (NONO) as highly enriched around circRNAs compared to other exons. Both proteins belong to the multifunctional *Drosophila* behavior/human splicing (DBHS) family with highly conserved RNA recognition motifs (RRMs) (*Dong et al., 1993*) and they are often found as a heterodimeric complex (*Knott et al., 2016*; *Knott et al., 2015*; *Lee et al., 2015*; *Passon et al., 2012*). The proteins are predominantly located to the nucleus, in particular to the membrane-less condensates known as paraspeckles (*Clemson et al., 2009*; *Fox et al., 2018*), where they play a pivotal role in cellular mechanisms ranging from regulation of transcription by interaction with the C-terminal domain (CTD) of RNA polymerase II (*Buxadé et al., 2008*; *Rosonina et al., 2005*; *Urban et al., 2000*), pre-mRNA splicing (*Emili et al., 2002*; *Ito et al., 2008*; *Kameoka et al., 2004*; *Peng et al., 2002*) and 3'end processing (*Kaneko et al., 2007*; *Rosonina et al., 2005*) to nuclear retention (*Zhang and Carmichael, 2001*) and nuclear export of RNA (*Furukawa et al., 2015*). Recently, SFPQ has been implicated in ensuring proper transcription elongation of neuronal genes (*Takeuchi et al., 2018*) representing an interesting link to circRNAs, as these are highly abundant in neuronal tissues and often derive from neuronal genes (*Rybak-Wolf et al., 2015*).

Here, we show that SFPQ depletion leads to specific deregulation of circRNAs with long flanking introns devoid of proximal inverted *Alu* elements. Moreover, we show that long introns in particular are prone to intron retention and alternative splicing with concomitant premature termination. While premature termination is not the main driver of circRNA deregulation, we provide evidence for a complex interplay between upstream (acting positively on circRNA production) and downstream features (acting negatively) that collectively govern the production of individual circRNAs in the absence of SFPQ. This not only elucidates a conserved role for SFPQ in circRNA regulation but also identifies upstream alternative splicing as an approach toward circRNA production.

## Results

### The DALI circRNAs are defined by long flanking introns and distal inverted *Alu* elements

To stratify circRNAs by their inverted *Alu* element dependencies, we characterized the circRNAome in two of the main ENCODE cell lines, HepG2 and K562 (*Supplementary file 1*). Using the joint prediction of two circRNA detection algorithms, ciri2 and find_circ, we identified 3044 and 7656 circRNAs in HepG2 and K562, respectively. While proximal inverted *Alu* elements (IAEs) are important for the biogenesis of a subset of circRNAs (*Jeck et al., 2013*; *Ivanov et al., 2015*), we and others have shown that long flanking introns associate with circRNA loci, particularly for the conserved and abundant circRNAs (*Stagsted et al., 2019*; *Westholm et al., 2014*), and the biogenesis of this group of circRNA species is largely unresolved. To focus our analysis on the non-*Alu*, long intron fraction of circRNAs, we subgrouped circRNAs based on their IAE distance and flanking intron length using median distance and length as cutoffs (*Figure 1A–C*). We observed that these two features show interdependent distributions, where approximately 70% of the top1000 expressed circRNAs group as either Distal-Alu-Long-Intron (DALI) circRNAs or Proximal-Alu-Short-Intron (PASI) circRNAs (*Figure 1D*). Apart from long flanking introns and distal IAEs, DALI circRNAs show higher overall expression compared to PASI circRNAs, longer genomic lengths, but similar distribution of mature lengths (*Figure 1—figure supplement 1A–D*). Moreover, almost half of a previously characterized subgroup of circRNAs, the AUG circRNAs (*Stagsted et al., 2019*), derive from DALI circRNAs (*Figure 1—figure supplement 1B*), and interestingly, when filtering circRNAs for conservation (in mouse and human), 69–72% of conserved circRNAs are DALI circRNAs (*Figure 1E*). This finding suggests that the IAE-dependent biogenesis pathways may not be relevant for the most conserved and abundant circRNAs and that other factors must be involved.

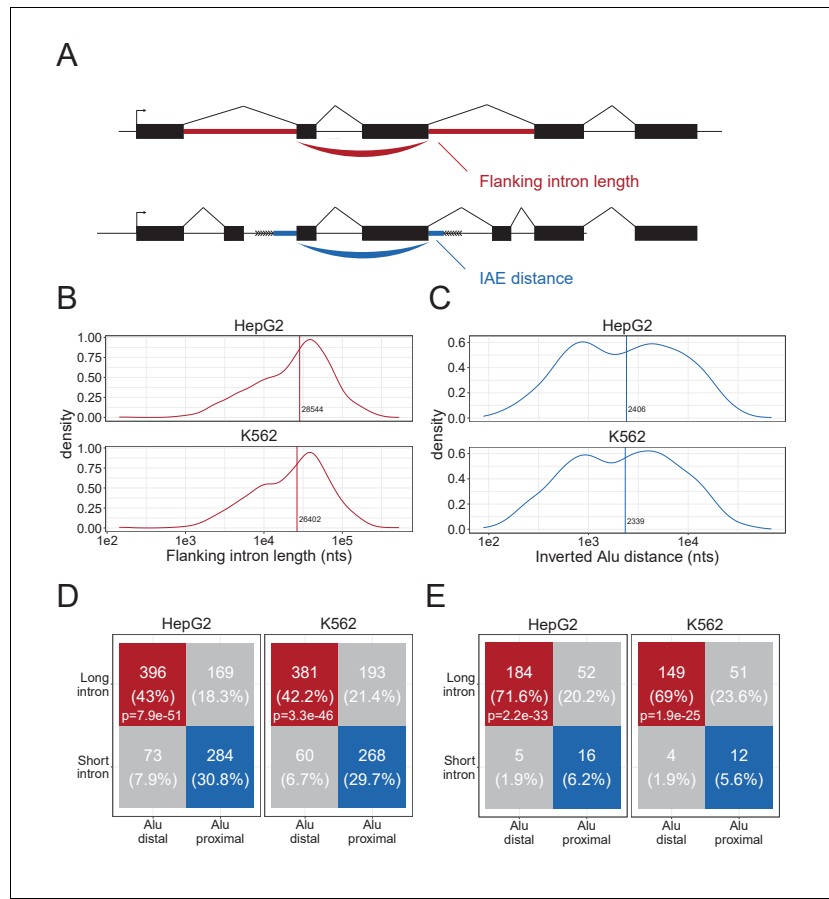

**Figure 1.** Characteristics of DALI-circRNA. (**A**) Schematics showing the flanking intron length (red) defined by the sum of annotated flanking introns and inverted *Alu* element (IAE) distance (blue) defined by the sum of distance to the most proximal IAE. (**B–C**) Density plot for the distribution of flanking intron lengths (**B**) and IAE Distance (**C**) for the top1000 expressed circRNAs in HepG2 (upper facet) and K562 (lower facet). The vertical line represents the median. (**D**) Contingency table showing the 4-way distribution of circRNAs with long and short flanking introns (in respect to the median) and proximal and distal IAEs (also in respect to the median, see B and C) for HepG2 (left facet) and K562 (right facet). The contingency table is color-coded by circRNA subgroup; DALI (distal *Alu*, long flanking introns, in red), PASI (proximal *Alu*, short flanking introns, in blue) and 'Other' (unclassified, in gray) circRNAs. The p-values are Fisher's exact test of independence. (**E**) As in D, but for the subset of circRNAs with conserved expression in mouse.

The online version of this article includes the following figure supplement(s) for figure 1:

**Figure supplement 1.** circRNAome in HepG2 and K562 from ENCODE.

## SFPQ and NONO are specifically enriched in the introns flanking DALI circRNAs

In order to identify RBPs that could drive circRNA formation, we used the elaborate ENCODE eCLIP data (*ENCODE Project Consortium, 2012*; *Supplementary file 2*). We scrutinized the immediate flanking regions of the 1000 most highly expressed circRNAs in HepG2 and K562 with the assumption that factors directly involved in backsplicing are likely to bind in the vicinity of the back-splicing sites. We extracted an eCLIP enrichment score using Wilcoxon rank-sum tests between the number of eCLIP reads aligned to circRNA flanking regions (upstream and downstream) compared to flanking regions of host exons, that is other exons from the circRNAs expressing genes. In HepG2, we found SFPQ to be the protein most highly enriched in the circRNA flanking regions, while NONO – a known interaction partner for SFPQ (*Dong et al., 1993*) – shows enrichment in K562 cells (*Figure 2A–B*, to our knowledge eCLIP datasets on SPFQ in K562 and NONO in HepG2 are not available). Comparing DALI and PASI circRNAs shows that SFPQ is DALI circRNA specific, both

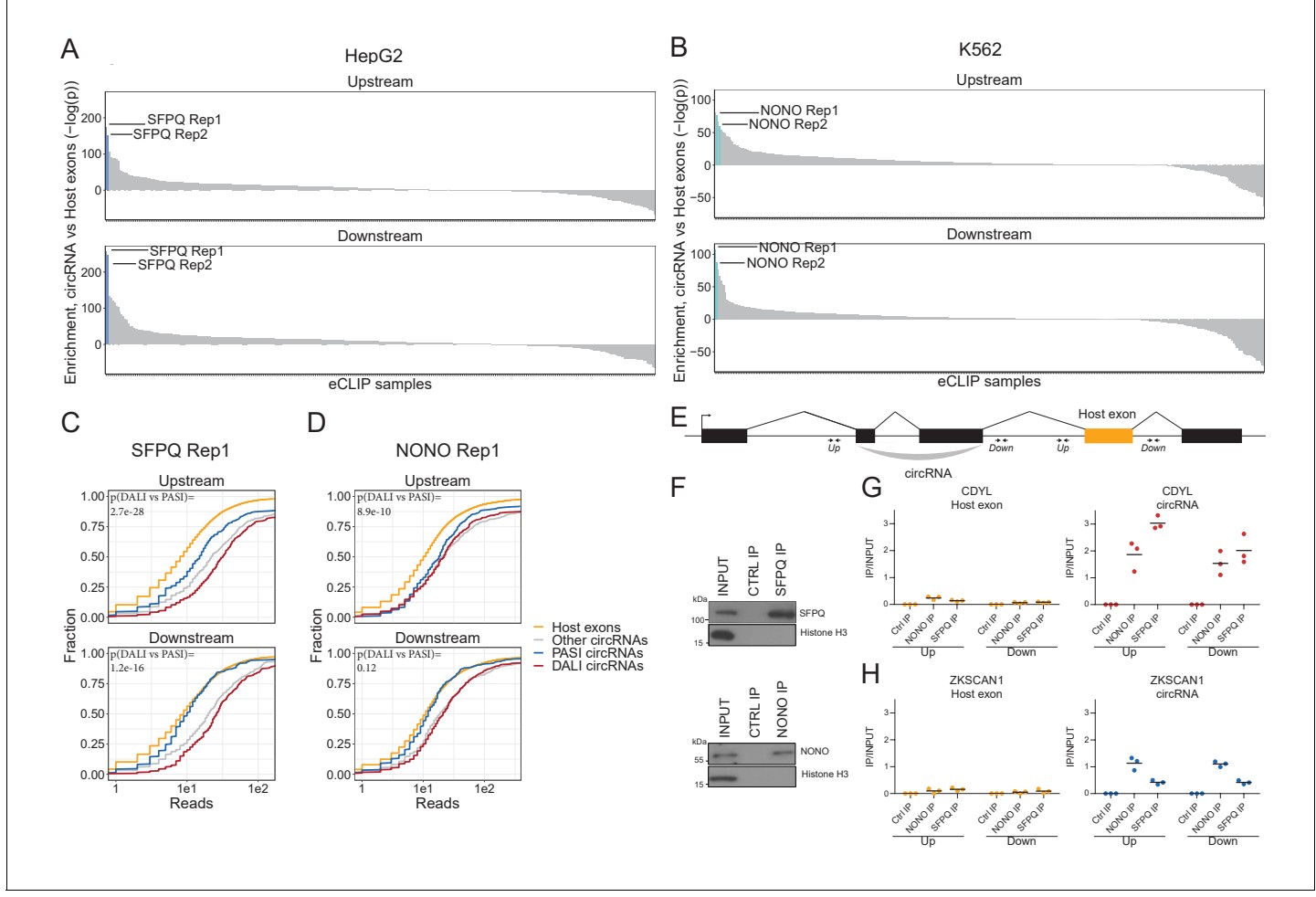

**Figure 2.** SFPQ and NONO show enriched binding in the flanking regions of DALI circRNAs. (A–B) Barplot showing enrichment/depletion of eCLIP signal (see *Supplementary file 2*) in the vicinity of circRNAs (+/- 2000 nt) compared to host exons (+/- 2000 nt) as determined by Wilcoxon rank-sum tests for HepG2 (A) and K562 (B) eCLIP samples. (C–D) Cumulative plots of SFPQ (C) and NONO (D) eCLIP read distribution upstream and downstream of circRNA subgroups and host exons as denoted. (E) Schematic showing localization of primers (+/- 2000 nt) for targeting either upstream (up) or downstream (down) intronic regions of splice sites in respect to circRNA exons or host exon. (F) Western blotting of immunoprecipitated (IP), endogenous SFPQ or NONO from nuclear fractions of HepG2 cells with Histone H3 as a loading control. (G–H) RT-qPCR of intronic regions flanking a downstream host gene exon (left facet) or flanking the circRNA producing exon(s) (right facet) of CDYL (G) and ZKSCAN1 (H) upon RNA IP of endogenous SFPQ or NONO from nuclear fractions of HepG2 cells. The relative expression of immunoprecipitate (IP)/input is plotted. Data for three biological replicates are shown.

The online version of this article includes the following figure supplement(s) for figure 2:

**Figure supplement 1.** SFPQ and NONO enriched on circRNA flanking introns.

**Figure supplement 2.** RNA immunoprecipitation of SFPQ and NONO confirms enrichment.

upstream and downstream of the circularizing exons (*Figure 2C*, p≤1.2e-16), whereas NONO associates with circRNA loci more generally and with the upstream regions of DALI circRNAs specifically (*Figure 2D*). SFPQ, like circRNAs, is known to associate with long introns (*Iida et al., 2020*; *Takeuchi et al., 2018*). To exclude that the enrichment seen is a mere bias from the flanking intron length, we extracted a subset of annotated splice acceptor (SA) and splice donor (SD) pairs sampled to match the expression level (linear spliced reads) and flanking intron lengths of DALI circRNAs (denoted 'DALI-like exons') (*Figure 2—figure supplement 1A–H*). This analysis shows that both SFPQ and NONO were significantly more enriched around circRNA exons compared to sampled DALI-like exons (*Figure 2—figure supplement 1E–H*).

To validate the binding of SFPQ and NONO on nascent circRNA transcripts, we conducted RNA immunoprecipitation (RIP) qPCR in HepG2 (*Figure 2E–H* and *Figure 2—figure supplement 2A–B*) and HEK293T cells (*Figure 2—figure supplement 2C–H*) and quantified the expression of a panel of representative DALI and PASI circRNAs. Here, the flanking regions of DALI circRNAs, circCDYL and circARHGAP5 (circEYA1 in HEK293T), were significantly enriched for SFPQ binding compared to downstream intronic regions (*Figure 2G* and *Figure 2—figure supplement 2A,E and G*). However, we found no enrichment for PASI circRNAs, circZKSCAN1 (*Figure 2H* and *Figure 2—figure supplement 2F*) and circNEIL3 (*Figure 2—figure supplement 2B and H*). Thus, we conclude that SFPQ and NONO associate with the flanking introns of DALI circRNAs, and this may be indicative of a functional role in circRNA biogenesis.

## SFPQ depletion represses DALI circRNAs production

To study the impact of SFPQ and NONO on circRNA production, we depleted SFPQ and NONO in HepG2 and HEK293T cells using two different siRNAs for each target (*Figure 3—figure supplement 1A*, *Supplementary files 3* and *4*). Western blot and RT-qPCR (*Figure 3A*, *Figure 3—figure supplement 1B–E*) showed that expression of both proteins was efficiently reduced upon siRNA treatment, although, unexpectedly, the expression levels of *SFPQ* mRNA appeared unaffected by SFPQ knockdown and greatly elevated upon NONO depletion (*Figure 3—figure supplement 1C and E*). This, we speculate, is the result of compensatory effects or autoregulatory mechanisms. We performed total RNA sequencing of the knockdown samples, and conducted gene expression analysis of circRNA and mRNAs. Principal Component Analysis (PCA) of HEK293T and HepG2 samples shows clear grouping of treatments (SFPQ, NONO, and CTRL), both on mRNA and circRNA levels (*Figure 3—figure supplement 1F–I*), suggesting that most of the variance between samples are explained by the knockdown. Although for HepG2, two samples (siSFPQ1_rep1 and siNONO2_rep2) display outlier signatures and were thus removed in downstream analyses. Overall, the composition and expression of DALI and PASI circRNAs in the HepG2 and HEK293T-derived samples look very similar to the ENCODE-based analysis (*Figure 3—figure supplement 1J–M*).

The differential circRNA expression analysis showed that DALI circRNAs are generally reduced upon SFPQ depletion, whereas PASI circRNAs are practically unaffected in both HEK293T (*Figure 3B and D*) and HepG2 (*Figure 3C and E*) cells. For NONO, we observed almost no impact on circRNA production in both cell lines (*Figure 3B–E*). This could either indicate that NONO is less involved in circRNA biogenesis, or that the effect is in part masked by the concomitant upregulation of SFPQ observed upon NONO depletion. Consistently, RT-qPCR analyses of abundant DALI circR-NAs, circCDYL (Figure 3F-G, *,Figure 3—figure supplement 2C*), circARHGAP5 (*Figure 3—figure supplement 2A*) and circEYA1 (*Figure 3—figure supplement 2E*), and PASI circRNAs, circZKSCAN1 (Figure 3H-I , *Figure 3—figure supplement 2D*) and circNEIL3 (*Figure 3—figure supplement 2B and F*) confirmed repressed expression of DALI circRNAs and unchanged PASI circRNAs expression relative to host gene levels. Finally, to support a direct role for SFPQ in circRNA formation, we overlaid the results from SFPQ-depleted HepG2 cells with the SFPQ eCLIP data and observed a significant association between SFPQ binding in the flanking regions of DALI circRNAs, as expected, but also a clear association with deregulated circRNAs compared to unchanged circRNAs (*Figure 3J*). Here, SFPQ appears to associate upstream and downstream of repressed circRNAs, whereas upregulated circRNAs only show significant enrichment in the upstream region. In addition, we examined previously published total RNAseq from SFPQ conditional knock-out (KO) mouse brain (*Takeuchi et al., 2018*; *Supplementary file 5*). Here, as in human cell lines, DALI and PASI circRNA are prevalent subclasses (*Figure 3—figure supplement 3A*) with DALI circRNAs showing higher abundancy compared to PASI circRNAs (*Figure 3—figure supplement 3B*). SFPQ depletion in mouse brain affects global circRNA expression (*Figure 3—figure supplement 3C*); however, in contrast to HEK293T and HepG2 cells, we found a more equal distribution of up- and downregulated circRNAs upon SFPQ removal (*Figure 3—figure supplement 3D–E*), and we detect a clear tendency for DALI circRNAs to be more prone to SFPQ-mediated regulation (25% vs 5% showing significant deregulation, *Figure 3—figure supplement 3F*, p=8.2e-80, Fisher's exact test). Consistent with HepG2, eCLIP analysis from mouse brain (*Takeuchi et al., 2018*) shows a clear association with DALI circRNAs, and a similar tendency toward upstream-only enrichment for circRNAs with increased expression upon SFPQ knockout (*Figure 3—figure supplement 3G*). Collectively, these findings

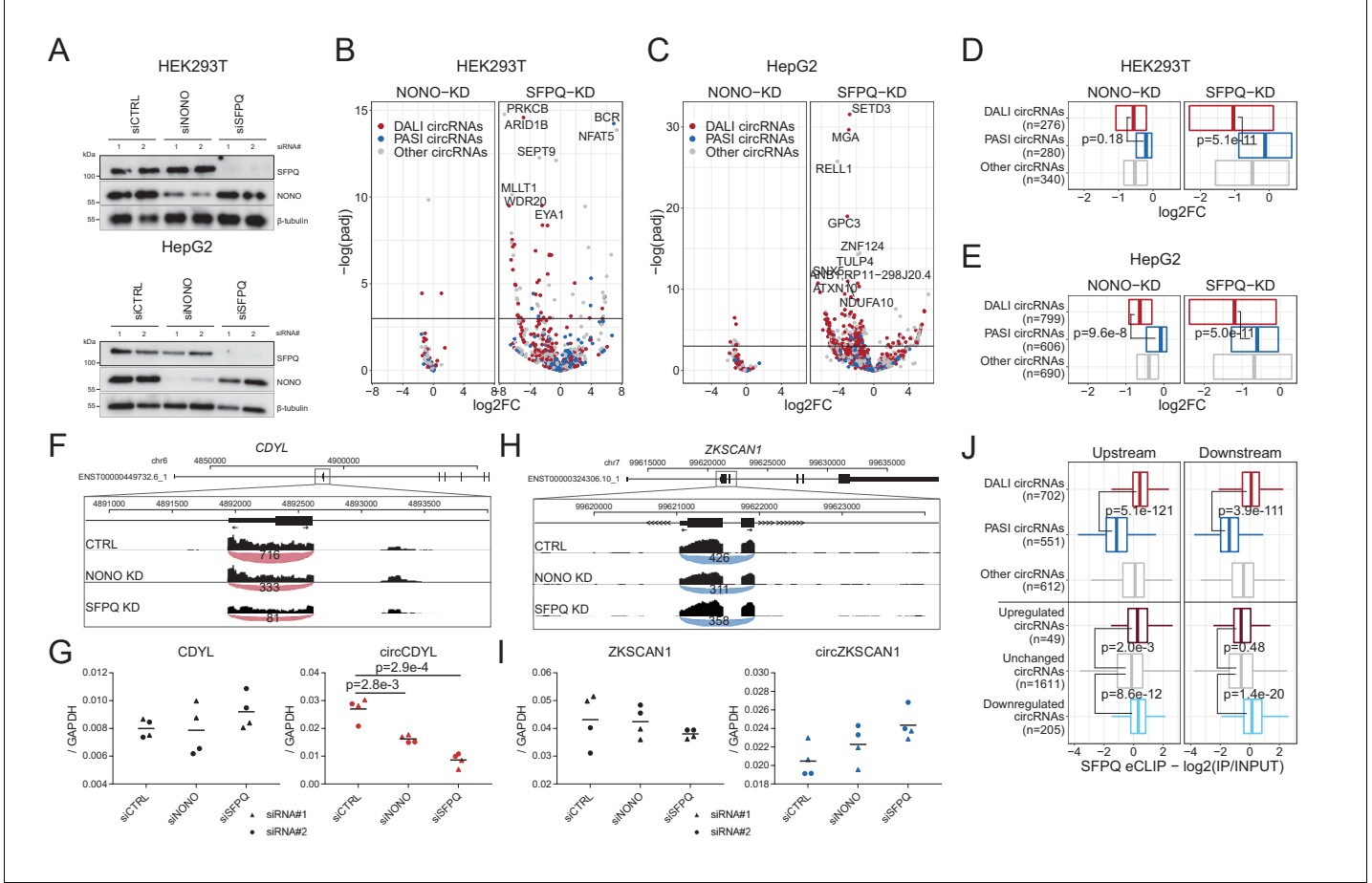

**Figure 3.** Knockdown of SFPQ affects DALI circRNAs. (**A**) Western blotting of proteins from HEK293T (upper panel) and HepG2 (lower panel) cells transfected with either CTRL siRNAs, siRNAs targeting NONO mRNA, or siRNAs targeting SFPQ mRNA using antibodies against SFPQ, NONO, and β-tubulin (loading control) as denoted. (**B–C**) Volcano plot showing deregulated circRNAs upon NONO (left facet) and SFPQ (right facet) depletion in HEK293T cells (**B**) or HepG2 cells (**C**) color-coded by circRNA subgroup; DALI circRNAs (red), PASI circRNAs (blue) and 'other' circRNAs (gray). (**D–E**) Boxplot showing overall changes in expression (log2Foldchange) of the three circRNA subgroups upon NONO and SFPQ depletion in HEK293T (**D**) and HepG2 (**E**) cells. p-Values are calculated using two-sided Wilcoxon rank-sum tests. (**F**) Genome screen dump of the circCDYL expressing locus with BSJ-spanning reads visualized as junction-track in the IGV browser (**G**) RT-qPCR quantification of circCDYL and linear CDYL expression upon SFPQ and NONO-depletion in HepG2 cells relative to *GAPDH* mRNA using two different siRNA designs for each target. Data for four biological replicates are shown. p-Values are calculated using Student's two-tailed t-test. (**H–I**) as in F and G, but for the PASI circRNA, circZKSCAN1. (**J**) Boxplot showing eCLIP enrichment for SFPQ either immediately upstream or downstream (within 2000 nucleotides from the circRNA splice sites) of expressed circRNAs stratified either by circRNA subgroup or by deregulation upon SFPQ depletion in HepG2 cells. p-Values are calculated using two-sided Wilcoxon rank-sum tests.

The online version of this article includes the following figure supplement(s) for figure 3:

**Figure supplement 1.** SFPQ/NONO-depletion in HEK293T and HepG2 cells.

**Figure supplement 2.** Expression profiles for selected circRNAs.

**Figure supplement 3.** CircRNAome analysis of SFPQ knockout mouse brain data (GSE60246).

suggest that SFPQ (and to a lesser degree NONO) regulates DALI circRNA biogenesis in mice and humans.

## SFPQ depletion affects alternative splicing and intron retention in long genes

Next, to understand the impact of SFPQ and NONO on transcription and splicing in general, we used the RNAseq data to investigate SFPQ/NONO-sensitive mRNAs. Here, we found that SFPQ-depletion triggers a general repression of long genes (stratified by median gene length, *Figure 4A*). The read distribution of highly repressed genes showed a peculiar expression profile with unaffected

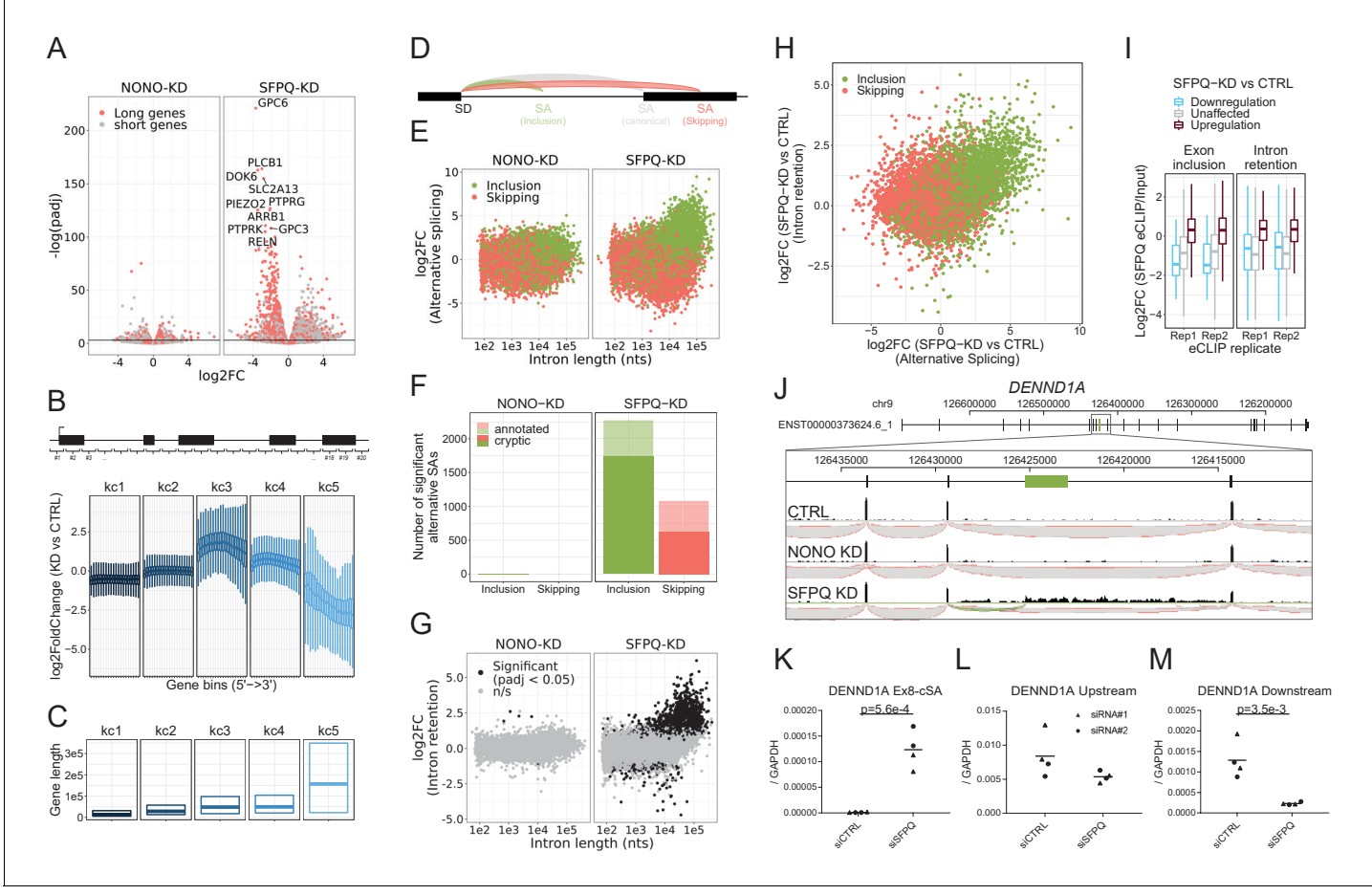

**Figure 4.** SFPQ ensures long-gene expression and suppresses cryptic splicing. (A) Volcano plot depicting differential expression of annotated genes upon NONO or SFPQ KD compared to CTRL in HepG2 cells, stratified by median gene length into 'long' and 'short' genes as denoted. (B) Boxplot showing binned expression of clustered genes. Each gene is sliced into 20 equally sized bins, and the differential expression of each bin is determined and subgrouped into five k-means clusters (kc) (see Materials and methods). (C) Boxplot showing gene lengths distribution (0.25, 0.5 and 0.75 quantiles) stratified by clusters obtained in B. (D) Schematic representation of alternative splicing, where canonical (gray) denoted the most abundant splicing from the splice donor in question. Inclusion (green) and skipping (red) denotes an alternative splicing event shorter or longer than canonical, respectively. (E) Scatter plot showing alternative splicing in NONO and SFPQ depleted samples as a function of canonical intron length and color-coded by type of splicing either inclusion or skipping, see schematics in D. (F) Barplot with the number of unique alternative splicing events showing significant deregulation upon NONO and SFPQ depletion stratified by inclusion (green) and skipping (red), and whether the alternative SA site is annotated (transparent) or not (opaque). (G) Scatter plot showing effects on intron retention (IR) upon SFPQ and NONO depletion as a function of intron length, color-coded by significance (adjusted p-value<0.05) as denoted. (H) Scatterplot showing for each detectable intron the correlation between changes in exon-inclusion/skipping (red/green) and intron retention upon SFPQ depletion. (I) Boxplot showing the IP/Input enrichment of SFPQ eCLIP reads in introns harboring an exon inclusion or an intron retention event color-coded by whether the event is up or down (red or blue, respectively) or not significant (n/s, gray). (J) Schematic showing coordinates and full genic locus of *DENND1A* (top panel) and exon 8 and 9 with alternative, unannotated exon in-between (green, middle panel). Merged intron-spanning reads (lower panel) from CTRL, NONO-KD, and SFPQ-KD samples (HepG2) are shown and color-coded by splicing type; canonical (gray), inclusion (green), and skipping (red), see D. (K–M) RT-qPCR analysis of alternative splicing event (K), upstream expression (L) and downstream expression (M) relative to *GAPDH* mRNA using two different siRNA designs for each target. Data for four biological replicates are shown. p-Values are calculated using student's two-tailed t-test.

The online version of this article includes the following figure supplement(s) for figure 4:

**Figure supplement 1.** Genic expression profile for selected long genes.

**Figure supplement 2.** SFPQ ensures long-gene expression (HEK293T + MOUSE).

**Figure supplement 3.** SFPQ co-expression rescue cryptic splicing.

read densities at the genic 5'ends but with dramatic reduction at the 3'end in HepG2 cells (*Figure 4—figure supplement 1A–D*) indicating that the transcription machinery drops off mid gene. This prompted us to survey genes globally for a 'drop-off' phenotype. Thus, we subgrouped genes into their expression profile by slicing each gene into 20 equally sized bins and conducting differential gene expression on all bins. To subgroup genes with of similar profiles in an unsupervised manner, we clustered the log2foldchanges across genes into five categories, denoted kc1-5, using k-means clustering (*Figure 4B*). Here, kc5 but also kc3 and 4 showed 'drop-off' effects but to different degrees, and interestingly, the effect correlates with gene length (*Figure 4B–C*). We obtain almost identical results from SFPQ-depleted HEK293T cells (*Figure 4—figure supplement 2A–C*) and mouse brain (*Figure 4—figure supplement 2F–H*).

Upon inspection of the downregulated genes in our samples, we found an upregulation of alternative splicing in the SFPQ KD samples (*Figure 4—figure supplement 1*). We classified all alternative splicing events as either inclusion or skipping relative to their respective canonical isoform (*Figure 4D*) and performed differential expression analysis using DESeq2. This showed an extensive change (mostly upregulation) of alternative splicing events correlating with intron length in both HepG2, HEK293T and mouse brain (*Figure 4E*, *Figure 4—figure supplement 2D and I*). Of the 2106 significantly deregulated inclusion events in HepG2, more than 96% are upregulated and of these, 76% are not annotated by gencode (*Figure 4F*, in HEK293T: 95% upregulated, 78% unannotated, in mouse: 90% upregulated, 88% unannotated: data not shown), and consequently, we suggest that these events are mostly cryptic or aberrant splicing. Furthermore, analyzing the levels of intron retention by quantifying unspliced intronic reads shows a very similar intron-length-dependent pattern with significant retention of long intron upon SFPQ depletion (*Figure 4G*, *Figure 4—figure supplement 2E and J*). Consistently, we find a clear correlation between exon inclusion and intron retention (*Figure 4H*), and a clear enrichment of SFPQ eCLIP signal in regions subjected to alternative splicing and intron retention (*Figure 4I*). As an example, for *DENND1A*, we observe a previously unannotated splicing event joining exon eight to an alternative splice acceptor dinucleotide (AG) residing in intron eight of this gene (*Figure 4J*), which is only detectable upon SFPQ knockdown (*Figure 4K*). In *DENND1A*, this cryptic event marks the transition from unaffected to repressed state, as quantification of the upstream region shows modest to no effect between control and knockdown, whereas the downstream region is highly suppressed (*Figure 4K–M*). To strengthen the direct effect of SFPQ on cryptic SA inclusion, we co-introduced an siRNA-resistant SFPQ expression vector (*Figure 4—figure supplement 3A–B*). This showed an almost complete rescue of the DENND1A cryptic splicing otherwise observed upon SFPQ depletion (*Figure 4—figure supplement 3C–E*).

Collectively, this suggests that intron retention and alternative splicing are conserved effects of SFPQ depletion, and that SFPQ plays a vital role in splicing integrity for long introns in particular.

## SFPQ depletion results in premature termination events

In order for alternative splicing to result in premature termination of transcription, the alternative/cryptic-included exons need to harbor a polyA-signal that can serve as a functional terminator of transcription. To investigate the magnitude of polyA-signal appearance in SFPQ knockdown samples, we subjected SFPQ and NONO-depleted HEK293T cells to 3'end quantSeq (*Figure 5—figure supplement 1A*, *Supplementary file 6*). Putative polyA-signals were retrieved using the MACS2 call-peak algorithm, and to further increase the signal-to-noise ratio, we characterized each peak by the presence of a bonafide polyA-signal (PAS: AAUAAA or AUUAAA). Furthermore, for each quantseq peak, we also extracted the highest prevalence of A's in all possible 15-nucleotide windows to reduce non-polyA-tail artefacts in the samples. The fraction of PAS-containing peaks dropped markedly when regions with 14 or 15 nucleotides A-stretches were found (*Figure 5—figure supplement 1B*), suggesting that these A-rich peaks are likely polyA-tail-independent artefacts and were thus removed from the analysis. The remaining peaks were classified as PAS sites, and for all PASs, the genic origin was annotated, and the differential usage was determined by DESeq2. This showed a clear enrichment on intronic PAS and a repression of exonic PAS usage upon SFPQ knockdown (*Figure 5A*). As before, NONO-depletion only showed a modest effect.

As an upstream termination impacts downstream elements, we determined the relative genic position of up- and downregulated PASs. This showed a clear and general 5'region tendency of upregulated vs downregulated intronic PASs (*Figure 5B*), suggesting that activation of upstream PASs may subsequently repress the usage of downstream PASs. In addition, activation of upstream

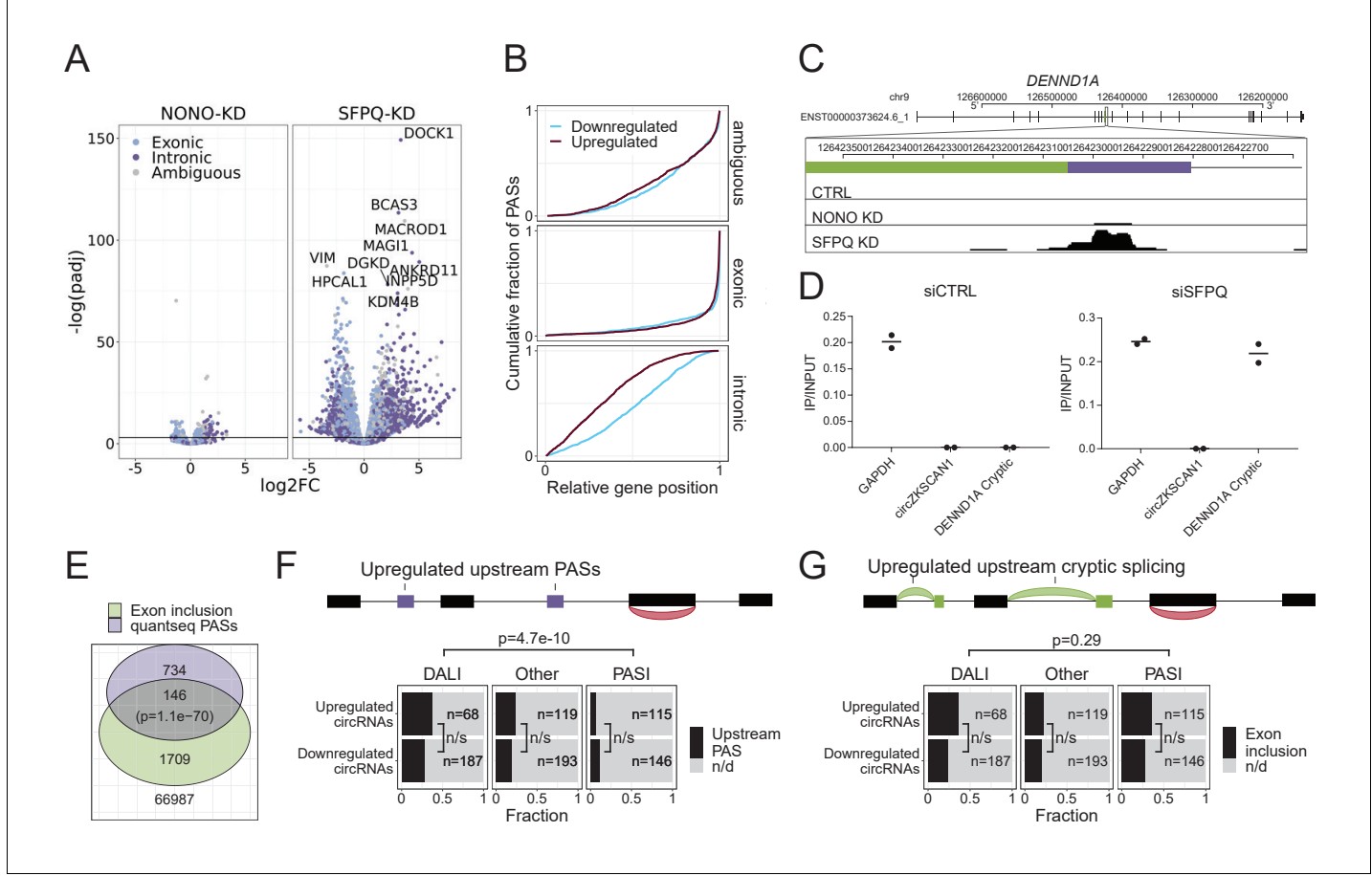

**Figure 5.** SFPQ depletion activates intronic polyA signal and premature termination. (**A**) Volcano plot showing deregulated PAS usage as measured by quantseq upon NONO and SFPQ depletion in HEK293T cells. PAS signals are color-coded by their genic origin; intronic (dark blue), exonic (light blue), or ambiguous (gray). (**B**) Plot showing the cumulative fraction of PASs as a function of relative genic position stratified by genic origin (ambiguous, exonic or intronic, vertical facets) and color-coded by whether the PAS is significantly up (red) or downregulated (blue) upon SFPQ knockdown. (**C**) Schematic representation of the DENND1A exon 8–9 locus with alternative exon (green) and putative PAS element (purple). Below, merged quantseq coverage from each experiment. (**D**) RT-qPCR on input and oligo-dT purified RNA from control and SFPQ-depleted HEK293T cells using amplicons specific for *GAPDH* mRNA (positive control), circZKSCAN1 (negative control), and the alternative SFPQ-activated exon. Values reflect ratios between oligo-dT purified and input quantities. Data for two biological replicates are shown. (**E**) Venn diagrams showing the number of unique introns with co-occurring upregulation of PAS and upregulated alternative splicing. The number of expressed introns without any evidence of enriched PASs or alternative splicing is denoted below the diagram. P-values are calculated by Fisher's exact test. (**F–G**) Schematic showing the outline of the analysis (upper panel): For each circRNA, the locus spanning from the promoter to the circRNA splice donor was interrogated for the presence of quantseq PASs (**F**) or exon inclusion (**G**). Barplot (lower panel) showing the fraction of upregulated and downregulated circRNAs upon SFPQ depletion in HEK293T cells with evidence of a concomitant upregulated upstream PAS (**F**) or an upstream exon inclusion event (**G**). Numbers indicate the total number of circRNAs in each group. p-Values are calculated by Fisher's exact test.

The online version of this article includes the following figure supplement(s) for figure 5:

**Figure supplement 1.** Quantseq analysis.

**Figure supplement 2.** U1 snRNA abundance upon SFPQ knockdown.

**Figure supplement 3.** circRNAs in kmeans clusters.

PASs were particularly pronounced in kc4 and 5 (*Figure 5—figure supplement 1C*) indicating that the 'drop-off'-phenotype may be a consequence of intronic PAS activation and premature termination. To investigate how alternative splicing relates to premature termination in a global manner, we assessed for the co-presence of alternative exon inclusion and significantly enriched PASs across all expressed introns (*Figure 5E*). Overall, this showed a significant overlap (*Figure 5E*) with kc4 exhibiting the highest degree of overlap with 72 distinct introns harboring both events (*Figure 5—figure supplement 1D*). For DENND1A, where cryptic splicing marks the transition from unaffected to

repressed state, we also observe a clear PAS with a consensus polyA signal (*Figure 5C*). This was validated using polyA enrichment, where the alternative transcript is oligo-dT purified as effectively as *GAPDH* mRNA only upon SFPQ knockdown (Figure 5D).

Collectively, this suggests that a notable fraction of genes exhibit alternative splicing and premature termination upon SFPQ knockdown with increased probability for longer introns, underscoring, once again, the importance of SFPQ in gene expression. Similarly, U1 snRNP has been shown to be important for repression of upstream, cryptic PAS usage in introns, known as U1 telescripting (*Gunderson et al., 1998*; *Kaida et al., 2010*; *Oh et al., 2017*). This raises the possibility that SFPQ could act in concert with U1 to protect long genes from premature transcriptional termination and polyadenylation. However, upon SFPQ knockdown U1 levels showed little or no response in HepG2 (*Figure 5—figure supplement 2A–B*) and HEK293T cells (*Figure 5—figure supplement 2C–D*), indicating that the SFPQ-dependent premature termination is independent from U1 telescripting.

## circRNA deregulation is not explained by premature termination

If SFPQ depletion results in wide-spread increase in premature termination, the observed deregulation of circRNAs in our dataset could simply be explained by incomplete transcription and not as a biogenesis effect per se. This notion is consistent with the fact that circRNAs in general and DALI circRNAs in particular associate with long flanking introns prone to alternative splicing and premature termination. However, not all circRNAs were depleted upon SFPQ knockdown and particularly in mouse brain, the DALI circRNAs were affected in both directions (i.e. up- and downregulated). To test whether the deregulation of circRNAs is driven by premature termination, we stratified circRNAs by their host gene clusters. This showed that while most circRNAs derive from kc1, 2, and 4, roughly the same expression profile is observed across all clusters (*Figure 5—figure supplement 3A–F*). In addition, comparing backsplicing to linear splicing from the circRNA producing loci, no clear correlation was observed, suggesting that the circRNA deregulation is not a mere consequence of transcription levels (*Figure 5—figure supplement 3G–I*). Finally, counting the prevalence of upstream (from the SD) significant intronic quantseq PASs (*Figure 5F*) or alternative splicing events (*Figure 5G*) there is no significant difference between up- and downregulated circRNAs, and for alternative splicing no difference between DALI and PASI circRNAs, whereas premature termination is more prominent upstream of DALI circRNAs. Collectively, we argue that premature termination is not the main driver of circRNA deregulation.

## Extracting features important for circRNA biogenesis

But what is then the underlying explanation for the deregulated expression of DALI circRNAs upon SFPQ depletion? As no single feature captures the circRNA deregulation accurately, we turned to multivariate regression analysis. Here, we collected a number of genic features (up- and downstream intron lengths, IAE distance, annotated distance to promoter and termination, and genomic length of circRNA), and differential expression data upon SFPQ depletion (linear up- and downstream splicing, flanking alternative splicing, upstream alternative splicing, up- and downstream intron retention) (*Figure 6A*). Pairwise correlation of all features shows modest redundancy but for certain combinations, such as 5′ linear splice (5′S) and 3′ linear splicing (3′S), we find a high level of positive interdependence (*Figure 6—figure supplements 1* and *2*), whereas intron retention generally correlates negatively with linear splicing (5′IR vs 5′S and 3′IR vs 3′S). In fact, linear splicing correlates negatively with all other features included in both HepG2 cells and mouse brain (*Figure 6—figure supplements 1* and *2*).

Splitting the quantified circRNAs into train and test sets (80:20 ratio), we trained a generalized linear model (GLM) against the observed circRNA log2foldchange. As all features were standardized, the resulting coefficients serve as a proxy for feature importance. Here, ranking features by coefficient, it is evident for both HepG2 and mouse brain that 5′ features generally correlate positively with circRNA production, whereas 3′ features (and IAE distance) correlate negatively (*Figure 6B and D*). As seen in both HepG2 and mouse brain, certain features, such as 5′IR (upstream intron retention), 5′ intron (upstream intron length), and 5′CSA (upstream cryptic SA usage), are highly distinctive for upregulated circRNA (*Figure 6C and E*), that is upstream aberrant splicing and intron retention stimulate circRNA production. Interestingly, correlating features show opposing effects on circRNA abundance. Here, according to the model, in HepG2, 5′S and 3′S impose positive and

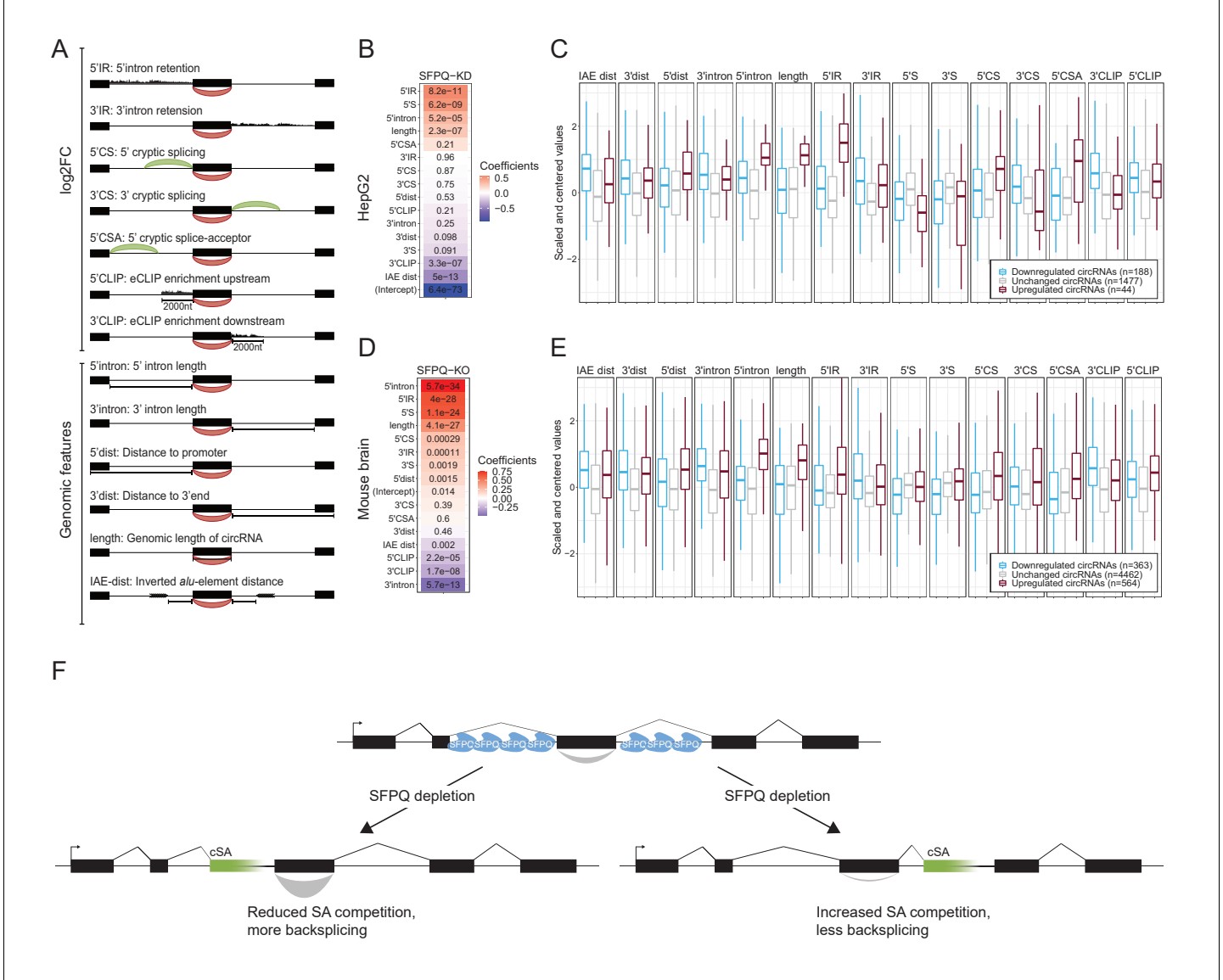

**Figure 6.** Multiple features contribute to circRNA regulation by SFPQ. (**A**) Schematic representation of features used in analysis. (**B**) Heatmap showing the feature coefficients from modeling circRNA deregulation (log2FoldChange) upon SFPQ depletion in HepG2 cells. The numbers within the heatmap are the associated p-values. (**C**) Boxplot showing the centered and scaled feature-values for significant up (red), significant down (blue), and unchanged (gray) circRNAs in HepG2. (**D–E**) as in B and C using mouse brain data. (**F**) Schematic depicting the SFPQ-mediated regulation of circRNA expression. Upon SFPQ knockdown, usage of cryptic splice acceptor sites (cSA) is induced, particularly within long introns. For upstream cSA inclusion (left scenario), the adjacent circRNA is upregulated possibly due to reduced competition with backsplicing, whereas for downstream cSA inclusion (right scenario), the circRNA is repressed due to increased competition with backsplicing,.

The online version of this article includes the following figure supplement(s) for figure 6:

**Figure supplement 1.** HepG2 features.

**Figure supplement 2.** Mouse brain features.

**Figure supplement 3.** HEK293T features and GLM model performance.

**Figure supplement 4.** GLM model performance.

negative impact on circRNA production, respectively (*Figure 6B*), whereas features showing anti-correlation (5'S and 5'IR) are both ascribed a positive coefficient in both HepG2 and mouse (*Figure 6B and D*). This was also observed in HEK293T, although less convincingly partly due to low sequencing depth in these samples (*Figure 6—figure supplement 3A–D*). Also, while the performance of the model on the test-set is modest but significant *Figure 6—figure supplement 4A and B*, Pearson

correlations: 0.41 (HepG2, p=1.8e-15) and 0.43 (mouse brain, p=1.8e-49), we observe convergence between the HepG2 and mouse brain-derived coefficients suggesting that the obtained features are conserved aspects of SFPQ-mediated circRNA regulation. Here, the most notable difference between HepG2 and mouse brain is the estimated intercept term (*Figure 6—figure supplement 4C*), which we interpret as the difference in cellular context, suggesting that the overall impact of SFPQ on circRNA expression may depend on other unidentified factors. Conclusively, we propose that aberrant SA usage caused by SFPQ depletion, most often observed in long introns, impacts transcription by invoking cryptic PAS usage and affects circRNA biogenesis by introducing competitive SAs. In case of cryptic SA inclusion immediately upstream a circRNA-producing locus, the cryptic SA engages with the upstream SD thus leaving the circRNA-dependent SA unattached and exposed for backsplicing. Alternatively, if the cryptic SA is located downstream, it may compete with backsplicing and thus repress circularization (*Figure 6F*).

## Discussion

The biogenesis of circRNAs is typically ascribed to the presence of proximal inverted repeat elements positioning the two splice sites involved in backsplicing into close proximity. For example, *Alu* elements are frequently found close to circRNA exons (*Ivanov et al., 2015*; *Jeck et al., 2013*), but since they are primate-specific (*Lander, 2011*), the biogenesis of most highly expressed and conserved circRNAs can not be explained by the presence of such elements (*Stagsted et al., 2019*). Additionally, by RNA association and dimerization, RBPs have a similar ability to juxtapose splice sites destined for backsplicing, although this currently only seems to apply to a few specific cases (*Conn et al., 2015*; *Errichelli et al., 2017*; *Ashwal-Fluss et al., 2014*; *Kramer et al., 2015*). Here, we attempted to further disclose the impact of RBPs on circRNAs biogenesis and to reveal features important for backsplicing. First, we subgrouped circRNAs into the *Alu*-dependent subset characterized by proximal IAEs and short flanking introns, termed PASI circRNAs, and the *Alu*-independent circRNAs with distal IAEs co-occurring with long flanking introns, the DALI circRNAs. By utilizing the extensive eCLIP resource made available by the ENCODE consortium, we identified SFPQ and NONO as two potentially interesting candidates, both associating significantly with the DALI circRNA producing loci. In both HepG2 and HEK293T cell lines as well as in mouse brain conditional knockouts, we observed a general deregulation of DALI circRNAs upon depletion of SFPQ and only subtle effects upon NONO KD, possibly due to the concomitant upregulation of SFPQ in these samples. Thus, we mostly focused our analysis on SFPQ. Here, apart from dramatic changes in DALI circRNA expression, we observed across all samples (HepG2, HEK293T, and mouse brain), that the absence of SFPQ results in aberrant splicing with extensive induction of cryptic splice acceptor sites, particularly in long introns. This correlates with a similar increase in intron retention, suggesting that these two phenotypes are closely coupled. Consistently, recent studies have shown SFPQ to associate with long introns (*Iida et al., 2020*; *Takeuchi et al., 2018*) and to be vital in regulating alternative splicing of long target genes ultimately affecting neural differentiation (*Luisier et al., 2018*) and axon development (*Thomas-Jinu et al., 2017*). Interestingly, circRNAs tend to originate from longer genes, especially neuronal genes (*Ragan et al., 2019*; *Rybak-Wolf et al., 2015*; *Szabo et al., 2015*; *You et al., 2015*), and exons prone to circularize are more frequently flanked by longer introns than non-circularized exons (*Jeck et al., 2013*), supporting SPFQ-sensitive regulation of circRNAs, in particular DALI circRNAs.

Using quantseq analysis, we found SFPQ-depletion to activate the use of intronic polyA-sites, which in many cases overlap with aberrant cryptic splicing thereby resulting in premature termination. Consistently, we also observe decreasing signal across the gene body in the absence of SFPQ. A model in which SFPQ facilitates the recruitment of CDK9 to the CTD of RNA polymerase II to maintain transcription elongation was recently proposed to explain the 'drop-off' effect seen upon SFPQ depletion (*Takeuchi et al., 2018*; *Hosokawa et al., 2019*). Instead, we claim that this is partly explained by the induced cryptic splicing and subsequent premature termination, emphasizing that transcription is highly coupled to splicing. SFPQ was initially found to be associated with the polypyrimidine tract, aid in the assembly of the spliceosome and be critical for the second catalytic step in splicing (*Ajuh et al., 2000*; *Makarov et al., 2002*; *Gozani et al., 1994*; *Patton et al., 1993*), supporting its role in splicing fidelity. Moreover, U1 snRNP abundancy has been described to be important for transcriptional elongation of long, neuronal genes (*Kaida et al., 2010*; *Berg et al., 2012*;

*Oh et al., 2017*). However, in many cases, the observed premature termination observed upon SFPQ depletion co-appears with an upstream cryptic SA. In addition, the U1 snRNA levels seem unaffected by SFPQ depletion suggesting that the SFPQ phenotype is not mediated entirely by U1 telescripting.

While DALI circRNAs in HepG2, HEK293T, and mouse brain are generally sensitive to SFPQ depletion, premature transcription termination fails to explain the observed circRNA levels. Instead, upstream intron length and aberrant splicing in the immediate upstream region have stimulating effects on circRNAs biogenesis, while, although less clear, cryptic events downstream show more detrimental impact. We speculate that SFPQ plays an imperative role in splicing fidelity, a role that becomes increasingly important with intron length. With the persistent presence of intronic sequences, cryptic and aberrant splicing become more likely, and cryptic exon inclusions in AU-rich introns will in many cases contain PAS-signal and thus cause premature termination. Consistent with this, RNA polymerase II has been shown to be stalled at AT-rich sequences (*Henriques et al., 2013*; *Palangat et al., 2004*), thus allowing a window of opportunity for splicing- and possible cleavage-directed transcription termination.

For circRNAs, the splice-sites involved in backsplicing must be protected from linear splicing as backsplicing occurs less effectively than canonical splicing. In particular, the SA has to remain unspliced until the RNA polymerase reaches the downstream SD. This can be facilitated by a fast polymerase elongation rate (*Zhang et al., 2016*) or by the lack of spliceosomal components (*Liang et al., 2017*). While SFPQ-depletion generally induces cryptic splicing imposing additional splice site competition, it also potentially eliminates upstream linear splicing and thus uncouples the SA from any upstream SD. This potentially exposes the upstream SA for backsplicing consistent with the observed positive impact of cryptic events (5' intron retention and 5' cryptic SA). In addition, we also observe that the mere length of the 5' intron has an important predictive value for circRNA formation. This, we hypothesize, is due to the high correlation with aberrant splicing and intron retention; both features are limited to detection in RNAseq. Supposedly, many of the cryptic events are not detectable in a steady state sequencing approach as they are likely unstable and subjected to nuclear quality control or nonsense mediated decay (NMD), and therefore intron length may serve as a useful proxy for aberrant splicing upon SFPQ-depletion. Furthermore, SFPQ is often described in various protein complexes with some comprising FUS and the nuclear resolvase DHX9. Like SFPQ, FUS has been shown to act in various processes within the cell, such as transcription regulation and RNA metabolism (*Lagier-Tourenne et al., 2012*; *Kwiatkowski et al., 2009*; *Vance et al., 2009*) but also to associate with the 5'ss of long introns (*Nakaya et al., 2013*; *Lagier-Tourenne et al., 2012*), especially those flanking circularizing exons, and hereby regulate circRNA biogenesis (*Errichelli et al., 2017*). DHX9 has, on the other hand, been shown to unwind intronic base pairing and thereby reduce the production of *Alu*-dependent circRNAs (*Aktaş et al., 2017*; *Errichelli et al., 2017*). Both proteins show interesting circRNA regulation abilities which could act cooperatively with SFPQ and thus affect the fate of DALI circRNAs upon SFPQ depletion.

Conclusively, we show that SFPQ is a key regulator of DALI circRNAs production by controlling and enforcing accurate long intron splicing. This highlights the complex and intricate relationship between splicing in general and backsplicing in particular. Furthermore, SFPQ has been associated to diverse neurological diseases, such as ALS (*Thomas-Jinu et al., 2017*; *Luisier et al., 2018*) and FTLD (*Ishigaki et al., 2017*), and may prove to be a critical for maintaining the circRNAome in these and other neurodegenerative pathologies. And while in steady state scenarios, cryptic splicing is negligible, it is interesting to speculate whether upstream cryptic splicing is generally involved in DALI circRNA production providing a useful tool for manipulating circRNA production without impacting host gene expression.

## Materials and methods

**Key resources table**

| Reagent type (species) or resource | Designation | Source or reference | Identifiers | Additional information |
|---|---|---|---|---|

*Continued on next page*

*Continued*

| Reagent type (species) or resource | Designation | Source or reference | Identifiers | Additional information |
|---|---|---|---|---|
| Cell line (*Homo sapiens*) | HEK293T | Invitrogen | N/A | |
| Cell line (*Homo sapiens*) | HepG2 | ATCC | RRID:CVCL_0027 | |
| Transfected construct (human) | pcDNA3- Myc-SFPQ | Genscript | This paper | Transfected construct |
| Transfected construct (human) | pcDNA3-Empty Vector | Invitrogen | N/A | Transfected construct |
| Antibody | Anti-SFPQ antibody (Polyclonal, Rabbit) | Abcam | RRID:AB_882523 | WB (1:5,000) |
| Antibody | Anti-nmt55/ p54nrb antibody (Polyclonal, Rabbit) | Abcam | RRID:AB_1269576 | WB (1:20,000) IP (2.5 µg) |
| Antibody | Anti-SFPQ antibody (Monoclonal, Mouse) | Sigma | RRID:AB_260995 | IP (2.5 µg) |
| Antibody | Anti-FLAG antibody (Monoclonal, Mouse) | Sigma | RRID:AB_262044 | IP (2.5 µg) |
| Antibody | Anti-beta Tubulin (Monoclonal, Mouse) | Millipore | RRID:AB_309885 | WB (1:2,000) |
| Antibody | Anti-MYC (Polyclonal, Rabbit) | Sigma | RRID:AB_439694 | WB (1:5,000) |
| Antibody | Anti-Histone H3 antibody (Monoclonal, Rabbit) | Abcam | RRID:AB_302613 | WB (1:1,000) |
| Antibody | Anti-Rabbit antibody (Polyclonal, Goat) | Dako | RRID:AB_2617138 | WB (1:5,000) |
| Antibody | Anti-Mouse antibody (Polyclonal, Goat) | Dako | RRID:AB_2617137 | WB (1:5,000) |
| Sequenced-based reagent | GAPDH FW | This paper | qPCR Primer | GTCAGCCGCATCTTCTTTTG |
| Sequenced-based reagent | GAPDH RE | This paper | qPCR Primer | GCGCCCAATACGACCAAATC |
| Sequenced-based reagent | SFPQ FW | This paper | qPCR Primer | ACAGGGAAAGGCATTGTTGA |
| Sequenced-based reagent | SFPQ RE | This paper | qPCR Primer | TCATCTAGTTGTTCAAGTGGTTCC |
| Sequenced-based reagent | NONO FW | This paper | qPCR Primer | TGATGAAGAGGGACTTCCAGA |
| Sequenced-based reagent | NONO RE | This paper | qPCR Primer | AGCGCATGGCATATTCATACT |
| Sequenced-based reagent | CDYL FW | This paper | qPCR Primer | ACCCACTAGTGCCTCAGGTG |
| Sequenced-based reagent | CDYL lin RE | This paper | qPCR Primer | ATTTCCTTTTGCTGGCAGTC |
| Sequenced-based reagent | CDYL circ RE | This paper | qPCR Primer | CTCGCTGTCATAGCCTTTCC |
| Sequenced-based reagent | ZKSCAN1 FW | This paper | qPCR Primer | CCCAGTCCCACTTCAAACAT |
| Sequenced-based reagent | ZKSCAN1 lin RE | This paper | qPCR Primer | TCCGCTGTGAATAGTGCAGA |
| Sequenced-based reagent | ZKSCAN1 circ RE | This paper | qPCR Primer | TCATTCAGGCTCCAGGAACT |
| Sequenced-based reagent | NEIL3 FW | This paper | qPCR Primer | CAGCCCAATACTCATCACCA |

*Continued on next page*

*Continued*

| Reagent type (species) or resource | Designation | Source or reference | Identifiers | Additional information |
|---|---|---|---|---|
| Sequenced-based reagent | NEIL3 lin RE | This paper | qPCR Primer | GAGGCGGTTGTGTTTACTGC |
| Sequenced-based reagent | NEIL3 circ RE | This paper | qPCR Primer | CGGGTACTTCATTAAGTGGCTAA |
| Sequenced-based reagent | EYA1 FW | This paper | qPCR Primer | CCAATGCCACTTACCAGCTT |
| Sequenced-based reagent | EYA1 lin RE | This paper | qPCR Primer | TACTGCTCCCAATTGCTGAA |
| Sequenced-based reagent | EYA1 circ RE | This paper | qPCR Primer | TTTCCCATCTGAACCTCGAC |
| Sequenced-based reagent | ARHGAP5 lin FW | This paper | qPCR Primer | CGTGTCAGCGGGAATAAAACT |
| Sequenced-based reagent | ARHGAP5 lin RE | This paper | qPCR Primer | TGGAATTAAAGGATCTGGCAGA |
| Sequenced-based reagent | ARHGAP5 circ Fw | This paper | qPCR Primer | CCTGCAATCACTTCTGACCA |
| Sequenced-based reagent | ARHGAP5 circ RE | This paper | qPCR Primer | TTTGGTTCTTTGTATTTCCCTCA |
| Sequenced-based reagent | DENND1A Upstream FW | This paper | qPCR Primer | CCAAGTTTTGTTTCCCCTTC |
| Sequenced-based reagent | DENND1A Upstream RE | This paper | qPCR Primer | AGAAGCAGCTCTTCGCTCCT |
| Sequenced-based reagent | DENND1A Ex8 FW | This paper | qPCR Primer | ACCAGAGAACTTCCCAGCAT |
| Sequenced-based reagent | DENND1A cSA RE | This paper | qPCR Primer | TGGGAGAGGGGAAATATGTG |
| Sequenced-based reagent | DENND1A Downstream FW | This paper | qPCR Primer | AAGAGCAGCTGCCAAAGACT |
| Sequenced-based reagent | DENND1A Downstream RE | This paper | qPCR Primer | GCGATGTTGCTCTTTGGTCT |
| Sequenced-based reagent | circARHGAP5 intronic Ups Fw | This paper | qPCR Primer | ATGGAATCATTGTGCTTTTC |
| Sequenced-based reagent | circARHGAP5 intronic Ups Re | This paper | qPCR Primer | AATCTTAATCTGGCCCAACTGA |
| Sequenced-based reagent | circARHGAP5 intronic Ds Fw | This paper | qPCR Primer | GGCTAAAAGCTGATTATTTGAAAAG |
| Sequenced-based reagent | circARHGAP5 intronic Ds Re | This paper | qPCR Primer | TACATTTTTCCAGGACTTTGTTCAT |
| Sequenced-based reagent | ARHGAP5 exon five intronic Ups Fw | This paper | qPCR Primer | TGTGGCTAAAACAGGGTGTG |
| Sequenced-based reagent | ARHGAP5 exon five intronic Ups Re | This paper | qPCR Primer | AGGCACCTACAACCAACAGC |
| Sequenced-based reagent | ARHGAP5 exon five intronic Ds Fw | This paper | qPCR Primer | AATGCTGGGTCACTTTGGTC |
| Sequenced-based reagent | ARHGAP5 exon five intronic Ds Re | This paper | qPCR Primer | CAGCCTGGTTCCTAACAAGC |
| Sequenced-based reagent | circCDYL intronic Ups Fw | This paper | qPCR Primer | TTTTGTCTTTGTTTAATGCCATTTC |
| Sequenced-based reagent | circCDYL intronic Ups Re | This paper | qPCR Primer | GGCCAGACTGAGTATACATAAGGAA |
| Sequenced-based reagent | circCDYL intronic Ds Fw | This paper | qPCR Primer | TGACCTGCAAGCTCAGAATGG |

*Continued on next page*

*Continued*

| Reagent type (species) or resource | Designation | Source or reference | Identifiers | Additional information |
|---|---|---|---|---|
| Sequenced-based reagent | circCDYL intronic Ds Re | This paper | qPCR Primer | GGATTGGTGGTGGAAGTAAAT |
| Sequenced-based reagent | CDYL exon seven intronic Ups Fw | This paper | qPCR Primer | CTGGTTCCTTGTGCCTTGAT |
| Sequenced-based reagent | CDYL exon seven intronic Ups Re | This paper | qPCR Primer | TTTTCAGGGAATGGAACTG |
| Sequenced-based reagent | CDYL exon seven intronic Ds Fw | This paper | qPCR Primer | CCTGCTCCTCACCTTCTCAC |
| Sequenced-based reagent | CDYL exon seven intronic Ds Re | This paper | qPCR Primer | GTGCTGCTTGTTCCTCTCCT |
| Sequenced-based reagent | circNEIL3 intronic Ups Fw | This paper | qPCR Primer | TTCGAGGCTGCAGTGAACTA |
| Sequenced-based reagent | circNEIL3 intronic Ups Re | This paper | qPCR Primer | TTGCCTTGTTCTTGTCTGGA |
| Sequenced-based reagent | circNEIL3 intronic Ds Fw | This paper | qPCR Primer | CGATCCAAGGTTGGTTGAAT |
| Sequenced-based reagent | circNEIL3 intronic Ds Re | This paper | qPCR Primer | TTTACACCAAATGGTCCCTCA |
| Sequenced-based reagent | NEIL3 exon five intronic Ups Fw | This paper | qPCR Primer | TACCCAAATCAGTAGGAATGAAGC |
| Sequenced-based reagent | NEIL3 exon five intronic Ups Re | This paper | qPCR Primer | CATACTGAACTCACGTGTTCCAA |
| Sequenced-based reagent | NEIL3 exon five intronic Ds Fw | This paper | qPCR Primer | AACCTGAGGGAGCCAAAGAT |
| Sequenced-based reagent | NEIL3 exon five intronic Ds Re | This paper | qPCR Primer | TGAAGCAGAGACTTTTGAAGG |
| Sequenced-based reagent | circZKSCAN1 intronic Ups Fw | This paper | qPCR Primer | ATGGCCAAGCTGGTCTTGAACTCC |
| Sequenced-based reagent | circZKSCAN1 intronic Ups Re | This paper | qPCR Primer | CAGGAACAGCTGTATGAAATGG |
| Sequenced-based reagent | circZKSCAN1 intronic Ds Fw | This paper | qPCR Primer | TGGAACACTTAACCATGACTGG |
| Sequenced-based reagent | circZKSCAN1 intronic Ds Re | This paper | qPCR Primer | CCATGCCTGGCTGATTTATTAT |
| Sequenced-based reagent | ZKSCAN1 exon four intronic Ups Fw | This paper | qPCR Primer | GCAACAGAGGGAGATGCTG |
| Sequenced-based reagent | ZKSCAN1 exon four intronic Ups Re | This paper | qPCR Primer | GTGTGTGCCAGGATCTTTGA |
| Sequenced-based reagent | ZKSCAN1 exon four intronic Ds Fw | This paper | qPCR Primer | GAAAACTCACAGAATTGGAGAAA |
| Sequenced-based reagent | ZKSCAN1 exon four intronic Ds Re | This paper | qPCR Primer | GGAGCCTTCAGAGGTCACAG |
| Sequenced-based reagent | circEYA1 intronic Ups Fw | This paper | qPCR Primer | CGGTCCATGGTTTTAAGAGTGA |
| Sequenced-based reagent | circEYA1 intronic Ups Re | This paper | qPCR Primer | TGCAACACAAGAAAGGCTGA |
| Sequenced-based reagent | circEYA1 intronic Ds Fw | This paper | qPCR Primer | AGCCTTGTTGTGGAGTAGCT |
| Sequenced-based reagent | circEYA1 intronic Ds Re | This paper | qPCR Primer | TCTTGTTTCCCATGCACACA |
| Sequenced-based reagent | EYA1 exon 12 intronic Ups Fw | This paper | qPCR Primer | CAGATTCTATTTTTGGCATGAGG |

*Continued on next page*

*Continued*

| Reagent type (species) or resource | Designation | Source or reference | Identifiers | Additional information |
|---|---|---|---|---|
| Sequenced-based reagent | EYA1 exon 12 intronic Ups Re | This paper | qPCR Primer | GGGCAAGTAAACAATTTCCAA |
| Sequenced-based reagent | EYA1 exon 12 intronic Ds Fw | This paper | qPCR Primer | CTCCCATCTCCCACCCTTTC |
| Sequenced-based reagent | EYA1 exon 12 intronic Ds Re | This paper | qPCR Primer | TCTCATCGAGCCTGGTTTGT |
| Sequenced-based reagent | U1 Fw | This paper | qPCR Primer | GCTTATCCATTGCACTCCGG |
| Sequenced-based reagent | U1 Re | This paper | qPCR Primer | CCCCACTACCACAAATTATGCA |
| Sequenced-based reagent | U1 | This paper | Northern blot Probe | ACAAATTATGCAGTCGAGTTT CCCACATTTGGGGAAAT CGCAGGGGTCAGCACATCCGGA |
| Sequenced-based reagent | 7SK | This paper | Northern blot Probe | TACTCGTATACCCTTGACCGAA GACCGGTCCTCCTCTATCGGGGATGGTC |
| Sequenced-based reagent | SFPQ siRNA #1 | Merck | SASI_Hs01_00073164 | Sense strand: GUACGAAUAUUCUCAGCGA[dT][dT] Antisense strand: UCGCUGAGAAUAUUCGUAC[dT][dT] |
| Sequenced-based reagent | SFPQ siRNA #2 | Merck | SASI_Hs01_00073165 | Sense strand: GGAAGAUGCCUAUCAUGAA[dT][dT] Antisense strand: UUCAUGAUAGGCAUCUUCC[dT][dT] |
| Sequenced-based reagent | NONO siRNA #1 | Merck | SASI_Hs02_00343478 | Sense strand: GAUGGAAGCUGCACGCCAU[dT][dT] Antisense strand: AUGGCGUGCAGCUUCCAUC[dT][dT] |
| Sequenced-based reagent | NONO siRNA #2 | Merck | SASI_Hs02_00343479 | Sense strand: CUCAGUAUGUGUCCAACGA[dT][dT] Antisense strand: UCGUUGGACACAUACUGAG[dT][dT] |
| Sequenced-based reagent | CTRL siRNA #1 | Merck | CAT#SIC001 | MISSION-siRNA Universal Negative Control #1 |
| Sequenced-based reagent | CTRL siRNA #2 | RiboTask | | Targets eGFP Sense strand: GACGUAAACGGCCACAAGUUC Antisense strand: ACUUGUGGCCGUUUACGUCGC |
| Commercial assay or kit | DNase I, RNase-free (1 U/μL) | Thermo Fisher Scientific | CAT# EN0521 | DNase Treatment |
| Commercial assay or kit | M-MLV Reverse Transcriptase kit | Thermo Fisher Scientific | CAT# 28025013 | Reverse trancription |
| Commercial assay or kit | NEBNext Poly(A) mRNA Magnetic Isolation Module | New England BioLabs Inc | CAT# E7490S | PolyA RNA selection |
| Commercial assay or kit | RiboCop rRNA Depletion Kit V1.2 | Lexogen | CAT# 037.24 | RNA ribodepletion |
| Commercial assay or kit | SENSE Total RNA-Seq Library Prep Kit | Lexogen | CAT# 009.24 | HEK293T RNA library preparation for total RNA sequencing |
| Commercial assay or kit | QuantSeq 3' mRNA-Seq Library Prep Kit | Lexogen | CAT# 038.24 | HEK293T RNA library preparation for 3'end RNA sequencing |
| Chemical compound, drug | Lipofectamine 2000 | Thermo Fisher Scientific | CAT# 12566014 | Transfection HEK293T and HepG2 |

*Continued on next page*

*Continued*

| Reagent type (species) or resource | Designation | Source or reference | Identifiers | Additional information |
|---|---|---|---|---|
| Chemical compound, drug | Lipofectamine RNAiMAX | Thermo Fisher Scientific | CAT# 13778150 | Transfection HepG2 |
| Chemical compound, drug | SiLentFect Lipid Reagent | Bio-Rad | CAT# 1703361 | Transfection HEK293T |
| Software, algorithm | GraphPad | Prism7 | RRID:SCR_002798 | |
| Software, algorithm | R | R Project for statistical computing | RRID:SCR_001905 | |
| Software, algorithm | Typhoon FLA 9500 | GE Healthcare | V. 1.1.0.187 | Northern blot |
| Software, algorithm | Image Studio | Licor Odyssey Fc | Ver 5.2 | Western blot |
| Software, algorithm | DESeq2, v1.24.0 | *Love et al., 2014* | RRID:SCR_015687 | |
| Software, algorithm | STAR, v2.7, | *Dobin et al., 2013* | N/A | |
| Software, algorithm | featureCounts, v2.0.0 | *Liao et al., 2014* | RRID:SCR_012919 | |
| Software, algorithm | ciri2 | *Gao et al., 2015* | N/A | |
| Software, algorithm | find_circ v1.2 | https://github.com/marvin-jens/find_circ | N/A | |
| Software, algorithm | annotate_circ.py | This paper, github/ncrnalab/pyutils | N/A | |
| Software, algorithm | RepeatMasker | UCSC Genome Browser | RRID:SCR_012954 | |
| Software, algorithm | liftOver tool | UCSC genome browser | RRID:SCR_018160 | |
| Software, algorithm | get_flanking spliced_reads.py | This paper, github/ncrnalab/pyutils | N/A | |
| Software, algorithm | get_spliced_reads.py | This paper, github/ncrnalab/pyutils | N/A | |
| Software, algorithm | get_alternative_splicing.py | This paper, github/ncrnalab/pyutils | N/A | |
| Software, algorithm | MACS2 peakcall, v2.2.6 | https://github.com/macs3-project/MACS | RRID:SCR_013291 | |
| Other | Dynabeads Protein A | Thermo Fisher Scientific | CAT# 10001D | RIP |
| Other | Dynabeads Protein G | Thermo Fisher Scientific | CAT# 10003D | RIP |
| Other | TRIzol Reagent | Thermo Fisher Scientific | CAT# 15596018 | RNA Extraction |
| Other | Platinum SYBR Green I Master kit | Thermo Fisher Scientific | CAT# 11733046 | qPCR assay |
| Other | 10% Tris-Glycine SDS-PAGE gel | Thermo Fisher Scientific | CAT# XP00102BOX | Western blot |
| Other | Immobilon-P Transfer Membrane | EMD Millipore | CAT# IPVH85R | Western blot |
| Other | SuperSignal West Femto Maximum Sensitivity Substrate kit | Thermo Fisher Scientific | CAT# 34095 | Western blot |
| Other | Amersham Hyperfilm ECL | GE Healthcare | N/A | Western blot |
| Other | Medical film | MG-SR plus, Konica Minolta | N/A | Western blot |

## Cell lines and transfections

HEK293T cells (Invitrogen) and HepG2 (ATCC, HB-8065) were cultured in Dulbecco's modified Eagle's media (DMEM) with GlutaMAX (Thermo Fisher Scientific) supplemented with 10% foetal bovine serum (FBS) and 1% penicillin/streptomycin sulphate (P/S). All cells were kept at 37℃ in a humidified chamber with 5% $CO_2$. Knockdown of SFPQ and NONO were carried out using transient transfections of siRNAs accordingly (siRNA sequences in *Supplementary file 7*): For HEK293T, approximately 250,000 cells were plated in a six-well dish and 24 hr later transfected with a final

concentration of 22.5 nM siRNA using siLentFect Lipid Reagent (Bio-Rad) accordingly to manufacturer's protocol. Forty-eight hr post-transfection, the cells were replenished with new media and re-transfected using Lipofectamine 2000 (Thermo Fisher Scientific). After additional 48 hr, cells were harvested. For HepG2, approximately 400,000 cells were plated in a six-well dish and reverse transfected with a final siRNA concentration of 50 nM using Lipofectamine RNAiMax (Thermo Fisher Scientific). Forty-eight hr post transfection, the cells were trypsinized before reverse transfected for the second hit using Lipofectamine 2000 (Thermo Fisher Scientific). After 48 hr, the cells were harvested (see *Figure 3—figure supplement 1A* for experimental outline). For the co-expression of EV or SFPQ WT upon CTRL or SFPQ KD, approximately 400,000 HEK293T cells were plated in six-well dishes. After 24 hr, the cells were transfected with a final concentration of 22.5 nM siRNA and 2.5 µg plasmid using Lipofectamine 2000 (Thermo Fisher Scientific) according to manufacturer's protocol, and the cells were harvested 48 hr post transfection. For all experiments, cells were harvested by washing in 1xPBS and subsequent centrifugation at 1200 rpm at 4˚C for 4 min. 66.6% of the harvested cells was used for RNA isolation, which was carried out using TRIzol Reagent (Thermo Fisher Scientific) according to manufacturer's protocol. Except for RNA used for RNAseq and RIP, 1 µg RNA was subjected to DNase I treatment (Thermo Fisher Scientific #EN0521) prior to subsequent analysis. The remaining cells (33.3%) were used for protein isolation; after centrifugation, the cell pellets were resuspended in 2xSDS loading buffer (125 mM Tris–HCl pH 6.8, 20% glycerol, 5% SDS, and 0.2 M DTT) and boiled at 95˚C for 5 min.

## RT-PCR and RT-qPCR

One µg of DNase-treated total RNA was reverse transcribed using the M-MLV Reverse Transcriptase kit (Thermo Fisher Scientific) according to manufacturer's protocol with the use of random hexamers to prime the reaction. In case of RT-PCR, the reaction was conducted with 30 cycles of PCR with or without RT enzyme (Primers listed in *Supplementary file 7*). The products were visualized by 1% agarose gel electrophoresis and verified using Sanger sequencing. For quantitative PCR, cDNA was mixed with Platinum SYBR Green I Master kit (Invitrogen) and ran on Light cycler 480 II instrument (Roche). The reactions were carried out in technical triplicates. The obtained Ct values for each triplicate were transformed (2-Ct) and averaged (σ). All samples were normalized to GAPDH. The results were visualized using GraphPad (Prism 7) with individual biological replicates are shown and the mean is plotted as a bar. For statistical analysis, Student's two-tailed t-test was used. p-Values below 0.05 (p<0.05) were considered significant. All statistical analyses were performed in GraphPad Prism.

## Poly(A) enrichment

Poly(A) enrichment was performed using NEBNext Poly(A) mRNA Magnetic Isolation Module (New England BioLabs Inc) according to manufacturer's protocol with five µg total RNA from CTRL-KD or SFPQ-KD in HEK293T cells used as input.

## RNA sequencing

For total RNA sequencing of HEK293T, RNA from SFPQ, NONO, and CTRL KD (using two different siRNAs for each condition with biological duplicates) were rRNA depleted using RiboCop rRNA Depletion Kit V1.2 (Lexogen) according to manufacturer's protocol. Subsequent cDNA libraries were prepared using SENSE Total RNA-Seq Library Prep Kit (Lexogen) following manufacturer's protocol.

For 3'end sequencing, cDNA libraries from HEK293T were prepared using QuantSeq 3' mRNA-Seq Library Prep Kit (Lexogen). For both methods, RNA quality was determined using the BioAnalyzer RNA nanochip (Agilent) and library concentration was quantified with KAPA Library Quant KIT RT-qPCR (Roche). Total RNAseq was done as 100nt paired-end sequencing and performed using the Illumnia platform (HiSEQ4000, BGI, Copenhagen), while for 3'end sequencing, 75nt single-end sequencing was performed at MOMA (Aarhus University Hospital) on a NextSeq500. For total RNA sequencing of HepG2 cells, library preparation and sequencing was performed at BGI (Copenhagen) using BGIseq.

## RNA-immunoprecipitation (RIP)

RIP was performed as previously described (*Rinn et al., 2007*) with some modifications to immunoprecipitate endogenous SFPQ and NONO. HepG2 and HEK293T cells were grown to confluence in 15 cm$^2$ dishes. Cells were harvested by trypsinization and resuspended in 2 ml PBS, 2 ml nuclear isolation buffer (1.28 M sucrose; 40 mM Tris-HCl Ph 7.5; 20 mM MgCl2; 4% Triton X-100) and 6 ml water on ice for 20 min (with frequent mixing). Nuclei were pelleted by centrifugation at 2,500G for 15 min. Nuclear pellet was resuspended in 1 ml RIP buffer (150 mM KCl, 25 mM Tris-HCl, pH 7.4, 5 mM EDTA, 0.5% Triton X-100) and 5 mM dithiothreitol (DTT) supplemented with Ribolock (Thermo Fisher Scientific) and proteinase inhibitor cocktail (Roche). Resuspended nuclei were split into two fractions of 500 µl each (for Mock and IP). Nuclear membrane and debris were pelleted by centrifugation at 13,000 RPM for 20 min. Antibody to SFPQ (P2860 Sigma), NONO (ab70335 Abcam), or FLAG epitope (Mock IP, F1804 Sigma) was added to supernatant (2.5 µg) and incubated for 4 hr at 4℃ with gentle rotation. Of protein A/G beads, 20 µl were added and incubated for 1 hr at 4℃ with gentle rotation. Beads isolated using magnetic, the supernatant were removed and beads were resuspended in 500 µl RIP buffer and repeated for a total of 5 RIP washes. Beads were divided into two fractions for protein (30%) and RNA (70%). Protein fraction was resuspended in 2xSDS loading buffer (125 mM Tris–HCl pH 6.8, 20% glycerol, 5% SDS, and 0.2 M DTT) and boiled at 95℃ for 5 min. RNA fraction was resuspended in 1 ml TRIzol Reagent (Thermo Fisher Scientific).

## Western blotting

Cells were harvested in 1xPBS and centrifuged at 1200 rpm at 4℃ for 5 min. For cell lysis, the cell pellet was collected and resuspended in 2xSDS loading buffer [125 mM Tris–HCl pH 6.8, 20% glycerol, 5% SDS, and 0.2 M DTT] and briefly boiled at 95℃ for 5 min before loading 1% on a 10% Tris-Glycine SDS-PAGE gel (Thermo Fisher Scientific) and run for app. 1.5 hr at 125 V. The proteins were transferred to an Immobilon-P Transfer Membrane (EMD Millipore) by wet-blotting ON at 4℃ at 25 V. Subsequently, the membrane was pre-blocked for 1 hr at RT with 10% skim milk, followed by 1 hr incubation with primary antibody (*Supplementary file 7*) and 1 hr with secondary antibody. After each antibody incubation, the membrane was rinsed 3 × 5 min in 1xPBS+0.05% Tween20 and 1 × 5 min wash with 1xPBS. The protein bands were developed using SuperSignal West Femto Maximum Sensitivity Substrate kit (Thermo Fisher Scientific) and Amersham Hyperfilm ECL (GE Healthcare) or Medical film (MG-SR plus, Konica Minolta).

## Mapping, circRNA detection, and quantification

Reads were mapped onto hg19 and mm10 for human and mouse data, respectively, with STAR (v2.7, *Dobin et al., 2013*) and quantified with featureCounts (v2.0.0, *Liao et al., 2014*) [featureCounts –p –O –i gene_id –t exon] using gencode annotations (v28lift37 for hg19 and v12 for mm10).

CircRNAs were predicted and quantified using ciri2 (*Gao et al., 2015*) and find_circ v1.2 (https://github.com/marvin-jens/find_circ) adhering to default settings, and only the shared predictions with ciri2 quantification were kept for analysis. circRNAs were annotated using *annotate_circ.py* (python scripts used are available at github/ncrnalab/pyutils). Flanking intron lengths were based on the mean total distance to the flanking exons based on gencode annotation (in case of multiple annotated flanking introns, the mean length was used), and IAE distance is the shortest possible Alu-mediated inverted repeat distance based on RepeatMasker (UCSC Genome Browser). For mouse, the IAE-distance is the shortest distance involving B1, B2 or B4 elements possible. DALI and PASI circRNAs were classified based on the median flanking intron lengths and median IAE distance in the sample. If no flanking introns were annotated, the circRNA was classified as 'other'. Furthermore, circRNAs were classified as conserved if both splice sites coincide exactly with previously detected mouse circRNAs (*Stagsted et al., 2019*) converted to hg19 coordinates using the liftOver tool (UCSC genome browser). Flanking linear spliced reads from the circRNA producing loci were extracted using *get_flanking spliced_reads.py*.

## Cryptic/alternative splicing and intron retention

First, all spliced reads were extracted from bam-files using *get_spliced_reads.py* requiring at least an eight nucleotide match on each exon. Then, separately for each splice-donor and –acceptor, all possible conjoining splice sites were extracted and counted using *get_alternative_splicing.py*. For each

splice site, the most abundant splicing event across all samples was denoted as canonical, whereas all other splicing events from that particular splice site were either classified as 'inclusion' if shorter or 'skipping' if longer than the canonical.

Based on the output from alternative splicing, for each splice-site the intronic region of the shortest alternative event was quantified using featureCounts (as above) but with [–minOverlap 5] to avoid quantification of any overlapping spliced reads.

## eCLIP analysis

Pre-mapped eCLIP datasets were downloaded from encodeproject.org (hg19, see *Supplementary file 2*). Based on the top 1000 expressed circRNAs from HepG2 and K562 (see *Supplementary file 1*), all reads aligning within 2000 nt upstream of the circRNA splice-acceptors or within 2000 nts downstream the circRNA splice-donors were counted using featureCounts. The same analysis was performed on all other annotated exons from the circRNAs host genes excluding the first and last host gene exons as well as exons involved in backsplicing. Moreover, genome-wide, exon-pairs were subsampled from gencode annotations to match the distributions of flanking intron lengths and linear spliced reads of DALI circRNAs; DALI-like exons. Enrichment was assessed by Wilcoxon rank-sum test between the number of reads flanking circRNAs compared to host exons. In case of SFPQ eCLIP on deregulated circRNAs, the number of eCLIP reads flanking the expressed circRNAs (within 2000 nt upstream and 2000 nt downstream) were retrieved using featureCounts (as above). For each locus, read counts were normalized to library depth (total reads) and to deduce the IP/INPUT enrichment, the mean of the replicate IP samples were divided by the input sample count. For SFPQ eCLIP from mouse brain (GSE96081), IP and INPUT samples were mapped to the mm10 genome using STAR (default setting), and then processed similar to the ENCODE eCLIP.

## Quantseq analysis

Quantseq reads were mapped onto hg19 using STAR as described above. The resulting bam-files were merged and divided into plus and minus-strand alignments. Then, using MACS2 peakcall (v2.2.6, https://github.com/macs3-project/MACS) with parameters [–nomodel –shift 0 –g 2.9e9], quantseq peaks were extracted. Each peak was then subsequently quantified using featureCounts and analysed for the presence of polyA signal (A[AU]UAAA) and the presence of polyA-stretches within the locus or in the immediate flanking regions (+/- 50 nts). Based on gencode annotation, each peak was assigned as exonic, intronic, ambiguous, or intergenic.

## Differential gene expression

First, in all analyses, low-expressed entries defined by mean counts across all samples < 1 and expressed in less than three samples were discarded. Then, analysis of differential gene expression was performed using DESeq2 (v1.24.0, *Love et al., 2014*) using formula ~ treatment, where treatment denotes the knockdown/knockout target. For mRNA and circRNA expression, the raw counts were merged and analysed in bulk. For conditional analysis, such as circ vs linear, alternative vs canonical splicing, and intron-retention vs intron-splicing, raw counts for each type was combined in one expression matrix with the associated design formula: ~ treatment * type, where type denotes circular or linear splicing (in case of circ vs linear). The log2FoldChange and p-adjust values from the interaction-term (treatment:type) was used in subsequent analyses. For binned analysis of transcripts, each locus was sliced and re-annotated as 20 equally sized bins irrespective of exon-intron structure, and this was then used in the featureCounts quantification. After differential expression analysis by DESeq2, genes were subgrouped into five k-means clusters based on the DESeq2-derived log2foldchange of all 20 bins.

An adjusted p-value below 0.05 was considered significant. All statistics were conducted in R (v3.6.3) and visualizations were done in R using ggplot2 (v3.3.0) and GraphPad.

## PAGE northern blot

PAGE northern was conducted as previously described (*Hansen, 2018*). Briefly, three µg RNA were dissolved in 40 µL loading buffer (8 M urea, 20 mM EDTA, 1% xylen, 1% bromophenolblue) and separated on a 10% PAGE gel in 1 x TBE buffer. Then, the RNA was transferred to a Hybond N+ membrane (GE Healthcare) by electroblotting overnight (ON) in 0.5 x TBE at 4°C. The membrane was UV

crosslinked on both sides using 120 mJ/cm$^2$ and then pre-hybridized in Church buffer (0.158 M NaH2PO4, 0.342 M Na2HPO4, 7% SDS, 1 mM EDTA, 0.5% BSA, pH 7.5) at 37°C for 1 hr. For visualization, the membrane was probed with 5' radioactively labeled DNA oligonucleotides at 37°C ON and washed twice in washing buffer (2x SSC, 0.1% SDS) for 10 min at 25°C prior to exposure on a phosphoimager screen, and finally scanned using a Typhoon imager (Amersham).

## Data accessibility

Sequencing data has been deposited on GEO (accession no GSE157622), and scripts for RNAseq data processing are available at github: github.com/ncrnalab/pyutils.

## Acknowledgements

We would like to thank Dr. Anne Færch Nielsen and Professor Jørgen Kjems for constructive comments and critical reading of the manuscript. This work was supported by the Novo Nordisk Foundation (NNF16OC0019874 to TBH) and the circRTrain ITN network.

## Additional information

### Funding

| Funder | Grant reference number | Author |
|---|---|---|
| Novo Nordisk Fonden | NNF16OC0019874 | Thomas Birkballe Hansen |

The funders had no role in study design, data collection and interpretation, or the decision to submit the work for publication.

### Author contributions

Lotte Victoria Winther Stagsted, Conceptualization, Validation, Investigation, Visualization, Methodology, Writing - original draft, Writing - review and editing; Eoghan Thomas O'Leary, Validation, Investigation, Visualization, Methodology, Writing - original draft, Writing - review and editing; Karoline Kragh Ebbesen, Investigation, Methodology, Writing - review and editing; Thomas Birkballe Hansen, Conceptualization, Data curation, Supervision, Funding acquisition, Validation, Investigation, Visualization, Writing - original draft, Writing - review and editing

### Author ORCIDs

Lotte Victoria Winther Stagsted https://orcid.org/0000-0003-1527-527X
Eoghan Thomas O'Leary https://orcid.org/0000-0002-1879-4238
Karoline Kragh Ebbesen https://orcid.org/0000-0003-3442-8577
Thomas Birkballe Hansen https://orcid.org/0000-0002-7573-9657

### Decision letter and Author response

Decision letter https://doi.org/10.7554/eLife.63088.sa1
Author response https://doi.org/10.7554/eLife.63088.sa2

## Additional files

### Supplementary files

• Supplementary file 1. ENCODE RNAseq. List of ENCODE dataset accession numbers used in RNAseq analysis.

• Supplementary file 2. ENCODE eCLIP. List of ENCODE dataset accession numbers used in eCLIP analysis.

• Supplementary file 3. HEK293T RNAseq. Mapping statistics and annotated output from featureCounts, ciri2 and find_circ using RNAseq on HEK293T cells.

- Supplementary file 4. HepG2 RNAseq. Mapping statistics and annotated output from feature-Counts, ciri2 and find_circ using RNAseq on HepG2 cells.
- Supplementary file 5. Mouse brain RNAseq. Mapping statistics and annotated output from feature-Counts, ciri2 and find_circ using RNAseq on mouse brain (GSE60246).
- Supplementary file 6. HEK293T QuantSeq. Mapping statistics and output from MACS2 analysis using quantseq data on HEK293T cells.
- Supplementary file 7. Primers, probes, and antibodies.
- Transparent reporting form

### Data availability

Sequencing data has been deposited on GEO (accession no GSE157622).

The following dataset was generated:

| Author(s) | Year | Dataset title | Dataset URL | Database and Identifier |
|---|---|---|---|---|
| Stagsted L, O'leary E, Hansen T | 2020 | The RNA-binding protein SFPQ preserves long-intron splicing and regulates circRNA biogenesis | https://www.ncbi.nlm.nih.gov/geo/query/acc.cgi?acc=GSE157622 | NCBI Gene Expression Omnibus, GSE157622 |

The following previously published datasets were used:

| Author(s) | Year | Dataset title | Dataset URL | Database and Identifier |
|---|---|---|---|---|
| Takeuchi A, Iida K, Tsubota T, Hosokawa M | 2018 | The RNA-binding protein Sfpq regulates long neuronal genes in transcriptional elongation | https://www.ncbi.nlm.nih.gov/geo/query/acc.cgi?acc=GSE60246 | NCBI Gene Expression Omnibus, GSE60246 |
| Takeuchi A, Denawa M, Iida K, Hagiwara M | 2018 | The RNA-binding protein Sfpq regulates long neuronal genes in transcriptional elongation [SFPQ_CLIP-seq] | https://www.ncbi.nlm.nih.gov/geo/query/acc.cgi?acc=GSE96081 | NCBI Gene Expression Omnibus, GSE96081 |

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
