## [Decision Letter]

**Acceptance summary:**

We believe that your paper provides relevant mechanistic insights about splicing of long introns, the generation of circular RNAs and the interplay between these two processes.

**Decision letter after peer review:**

Thank you for submitting your article "The RNA-binding protein SFPQ preserves long-intron splicing and regulates circRNA biogenesis" for consideration by *eLife*. Your article has been reviewed by three peer reviewers, including Juan Valcárcel as the Reviewing Editor and Reviewer #1, and the evaluation has been overseen by James Manley as the Senior Editor.

The reviewers have discussed the reviews with one another and the Reviewing Editor has drafted this decision to help you prepare a revised submission.

Summary:

Some circRNAs are flanked by inverted repeat elements, but many are not and this paper investigates the factors that control this latter class of RNAs. The authors find, using ENCODE circRNA and eCLIP data, as well as their own RIP results, that SFPQ is highly enriched in the introns flanking these circRNAs. siRNA depletion of SFPQ or its interacting partner NONO indeed regulate circRNA levels, concomitant with activation of alternative/cryptic splice sites, upregulation of intron retention and premature polyadenylation. Multivariate regression analyses indicate that upstream intron length and aberrant splicing in the immediate upstream region impact on circRNA biogenesis. The authors propose that a combination of aberrant splicing, premature termination and other elements control the production of circRNAs of the DALI (Distal Alu Long Intron) type.

The manuscript provides new insights into circRNA biosynthesis and efficient splicing of long introns, showcases the complexity of the interplay between various mechanisms for proper RNA processing and provides original leads for the field to pursue. The manuscript however falls short of demonstrating direct effects of SFPQ on circRNA production and of proving the cause-effects of the interplay between RNA processing steps.

Essential revisions:

1) The authors should try to rescue the effects of the knock down by expressing wild type and mutant forms of SFPQ and thus rigorously prove that SFPQ is responsible for these activities and identify critical protein domains involved. They should also use minigenes to show that exon circularization depends on the presence of the SFPQ sites and test the contribution of intronic polyadenylation sites. SFPQ could help avoiding the use of premature cleavage and polyA sites, which are known to alter circRNA formation (Liang et al., 2017). The authors should discuss this in the context of the role of U1 snRNP on usage of intronic PAS (the telescripting mechanism uncovered by Dreyfuss and colleagues). The authors should also determine if U1 snRNA levels are altered as this may be part of the reason for the widespread transcription termination effects that are observed.

2) Removal of SFPQ has very different effects on circRNA levels in HepG2 and HEK293T cells (mostly downregulation of DALI circRNAs) compared to in mouse brain (equal distribution of up- and down-regulated circRNAs). The authors should address this, to start with by looking into the CLIP-seq data from Takeuchi et al., 2018 and explore the binding of SFPQ to the introns flanking the down-regulated vs. up-regulated circRNAs. They should acknowledge the previous work by Takeuchi et al., 2018 on the function of SFPQ in transcription and discuss the drop-off phenotype showing that mice loss of SFPQ impairs transcriptional elongation accompanied by a decrease in RNA Pol II density.

---

## [Author Response]

Essential revisions:1) The authors should try to rescue the effects of the knock down by expressing wild type and mutant forms of SFPQ and thus rigorously prove that SFPQ is responsible for these activities and identify critical protein domains involved.

We thank the reviewers for this suggestion. Regarding a SFPQ rescue experiment, we have pursued several approaches. First, we tried targeting SFPQ in the 3’UTR using four different siRNAs followed by co-introduction of a SFPQ WT expression plasmid. However, we were unable to obtain any successful depletion of endogenous SFPQ by this method. Then, we designed a Myc-tagged SFPQ WT expressing plasmid resistant to one of the functional siRNAs (SFPQ siRNA#1) used in this paper. Unfortunately, effective co-transfection of SFPQ WT in conjunction with a 5-day knockdown protocol was challenging due to low transfection efficiency. Instead, we established a short-term 2-day, one-hit rescue setup. Here, western blotting demonstrated successful KD of endogenous SFPQ and Myc-SFPQ WT expression in our two day, one hit KD protocol in HEK293T cells (now included as Figure 4—figure supplement 3). We observed a substantial rescue of our cryptic spliced transcript, DENND1A, supporting a direct role of SFPQ in regulating cryptic splicing. However, the two-day protocol is insufficient to examine the effects on DALI circRNA expression, due to their high level of stability.

Regarding rescue-experiments with SFPQ domain mutants, this is definitely an interesting future avenue of research, however, in the current manuscript, we have not focused on the individual domains of SFPQ and therefore we argue that this is currently beyond the scope of this paper.

They should also use minigenes to show that exon circularization depends on the presence of the SFPQ sites and test the contribution of intronic polyadenylation sites. SFPQ could help avoiding the use of premature cleavage and polyA sites, which are known to alter circRNA formation (Liang et al., 2017). The authors should discuss this in the context of the role of U1 snRNP on usage of intronic PAS (the telescripting mechanism uncovered by Dreyfuss and colleagues). The authors should also determine if U1 snRNA levels are altered as this may be part of the reason for the widespread transcription termination effects that are observed.

We agree that the use of minigenes to show that exon circularization depends on the presence of the SFPQ sites, if successful, would be an important addition to this study. While SFPQ is known to have enriched binding in long introns of target RNAs (Iida et al., 2020; Takeuchi et al., 2018), SFPQ lacks a defined RNA-binding consensus sequence (Iida et al., 2020; 1). This suggests that SFPQ binding specificities may be regulated by a more complex mechanism, where the recruitment mechanism, binding proportion, and co-binding partners may affect SFPQs roles in transcription, splicing or stabilization of pre-mRNAs. As a consequence of SFPQs undefined binding specificity we are unable to generate a mini-gene system where we could confidently establish that exon circularization is dependent on SFPQ binding.

Furthermore, we agree that SFPQ’s ability to preserve long gene transcription through repression of intronic PAS usage is similar to the U1 telescripting mechanism uncovered by Dreyfuss and colleagues.

As suggested by the reviewers, we have conducted northern blotting and qRT-PCR analysis to determine the effect of SFPQ depletion on U1 snRNA levels. Here, U1 snRNP levels were not found to be significantly affected by SFPQ knockdown for both HepG2 and HEK293T cells (new Figure 5—figure supplement 2 ). Based on these observations, we cannot exclude any changes in U1 occupancy or localization upon SFPQ depletion, and the relationship between SFPQ and U1 could prove to be an interesting direction to pursue further. However, our data suggest that the effect of SFPQ depletion on intronic PAS usage cannot solely be explained by U1 shortage. This is now also included in the Discussion:

“Moreover, U1 snRNP abundancy has been described to be important for transcriptional elongation of long, neuronal genes by protecting against aberrant PAS usage (Kaida et al., 2010; Berg et al., 2012; Oh et al., 2017). However, in many cases the observed premature termination upon SFPQ depletion co-appears with an upstream cryptic SA, and the U1 snRNA levels seem unaffected by SFPQ depletion suggesting that the SFPQ phenotype is not mediated entirely by U1 telescripting.”

2) Removal of SFPQ has very different effects on circRNA levels in HepG2 and HEK293T cells (mostly downregulation of DALI circRNAs) compared to in mouse brain (equal distribution of up- and down-regulated circRNAs). The authors should address this, to start with by looking into the CLIP-seq data from Takeuchi et al., 2018 and explore the binding of SFPQ to the introns flanking the down-regulated vs. up-regulated circRNAs. They should acknowledge the previous work by Takeuchi et al., 2018 on the function of SFPQ in transcription and discuss the drop-off phenotype showing that mice loss of SFPQ impairs transcriptional elongation accompanied by a decrease in RNA Pol II density.

We thank the reviewers for this suggestion, and we have now included the eCLIP data from Takeuchi (GSE96081), shown in Figure 3—figure supplement 3G. Consistent with the SFPQ eCLIP from HepG2 cells, DALI circRNAs in the mouse brain data show highly enriched SFPQ binding in the flanking regions. Furthermore, to better grasp the impact of SFPQ on circRNA biogenesis, we now stratify the eCLIP analysis by upstream and downstream binding. Interestingly, this shows that particularly for circRNA with increased expression upon SFPQ repression, SFPQ binds solely to the upstream region, whereas downregulated circRNAs generally show enriched SFPQ binding both up- and downstream (see Figure 3J and Figure 3—figure supplement 3G). Additionally, we also now include the eCLIP data in the final GLM model. Here, most particularly for eCLIP signal in the downstream region, enrichment correlates negatively with circRNA expression upon SFPQ depletion, in mouse and HepG2, emphasizing that SFPQ binding in the flanking regions (mostly downstream) support circRNA expression (see also the schematic model in Figure 6F).

Reference:

1) Van Nostrand EL, Freese P, Pratt GA, Wang X, Wei X, Xiao R, et al. A large-scale binding and functional map of human RNA-binding proteins. Nature. 2020 Jul 29;583(7818):711–9.